# Distinct evolutionary trajectories of two integration centres, the central complex and mushroom bodies, across Heliconiini butterflies

Max S Farnworth[1]*, Yi Peng Toh[2], Theodora Loupasaki[1], Elizabeth A Hodge[1], Basil el Jundi[3,4], Stephen H Montgomery[1]*

[1]School of Biological Sciences, University of Bristol, Bristol, United Kingdom; [2]Behaviour and Speciation, Faculty of Biology, Ludwig-Maximilians-Universität München, Munich, Germany; [3]Department of Biology, Norwegian University of Science and Technology, Trondheim, Norway; [4]Institute of Biology and Environmental Sciences, Carl von Ossietzky Universität Oldenburg, Oldenburg, Germany

*For correspondence:
m.farnworth@bristol.ac.uk (MSF);
s.montgomery@bristol.ac.uk
(SHM)

**Competing interest:** The authors declare that no competing interests exist.

## eLife Assessment

The analysis of neural morphology across Heliconiini butterfly species revealed brain area-specific changes associated with new foraging behaviours. While the volume of the centre for learning and memory, the mushroom bodies, was known to vary widely across species, these new, **valuable** results show conservation of the volume of a center for navigation, the central complex, but with specific changes in neuropeptide expression in the noduli and in the numbers of ellipsoid body ring neurons. The presented evidence is **convincing** for both volumetric conservation in the central complex and fine neuroanatomical differences associated with pollen feeding, delivered by experimental approaches that are applicable to other insect species. This work will be of interest to evolutionary biologists, entomologists, and neuroscientists.

**Abstract** Neural circuits evolved to produce variable cognitive processes through adaptive mechanisms operating within a background of developmental and functional constraints. Understanding how this conflict is resolved requires a comparative framework encapsulating clear behavioural variation. We leverage Heliconiini butterflies to examine how selection shaped the evolution of the central complex and mushroom bodies, two insect integration centres involved in navigation. The evolution of systematic spatial foraging in *Heliconius* has led to changes in brain morphology and learning and memory profiles over a short evolutionary timescale. Here, we show that in contrast to massively expanded mushroom bodies, the central complex is strongly conserved in size and general architecture. However, we identify divergences in the expression of a neuropeptide, Allatostatin A, in the noduli, and in the numbers of GABA-ergic ring neurons and their branching in the fan-shaped body, which are essential members of the anterior compass pathway. These differences are rare examples of divergence inside the central complex network matching expectations of where evolutionary adaptability might occur. We conclude that due to the contrasting volumetric conservation of the central complex, and the massive differences in the mushroom bodies, their circuit logics must determine distinct responses to selection associated with divergent foraging behaviours.

## Introduction

Intricate patterns of interconnecting neurons in the brain are the anatomical basis of the processes that facilitate behaviour, be it an innate response to specific sensory cues or the integration of external cues with internal information (*Roberts et al., 2022*; *Jourjine and Hoekstra, 2021*). As evolutionary processes act on variation in behavioural traits and responses, local circuits must alter behaviour in the context of selection acting on the output of the larger system (*Farnworth and Montgomery, 2024b*). These alterations also occur in the context of a system's ability to respond to change (adaptability) which is in turn shaped by a range of potential constraining factors, be it to maintain ancestral or interrelated functions, or through partially shared deterministic developmental programs (*Montgomery et al., 2016b*; *Tosches, 2017*; *Hartenstein et al., 2021*).

A collection of neural circuits particularly suited to examine the interplay of constraints and adaptability are those that support experience-dependent spatial foraging behaviours in insects. Here, two well-studied brain regions, the mushroom bodies and the central complex, integrate multiple cues used for foraging such as visual landmarks, odour plumes as well as other directional cues to guide navigational behaviours (*Webb and Wystrach, 2016*; *Sun et al., 2020*; *Buehlmann et al., 2020*; *Collett and Collett, 2018*; *Seelig and Jayaraman, 2015*). The mushroom bodies, often referred to as the learning and memory centres of the insect brain, integrate multimodal sensory information with past experiences to modify a multitude of downstream connections according to learned and memorised contexts (*Li et al., 2020*; *Lin, 2023*). Similarly, the central complex, known as the navigation centre of the insect brain, relies on multisensory information (*El Jundi et al., 2015*; *Heinze and Reppert, 2011*; *Nguyen et al., 2022*) to produce steering signals by comparing the insect's current heading direction with its goal direction (*Stone et al., 2017*; *Beetz et al., 2023*; *Mussells Pires et al., 2024*; *Westeinde et al., 2024*). The central complex receives input from the mushroom bodies in

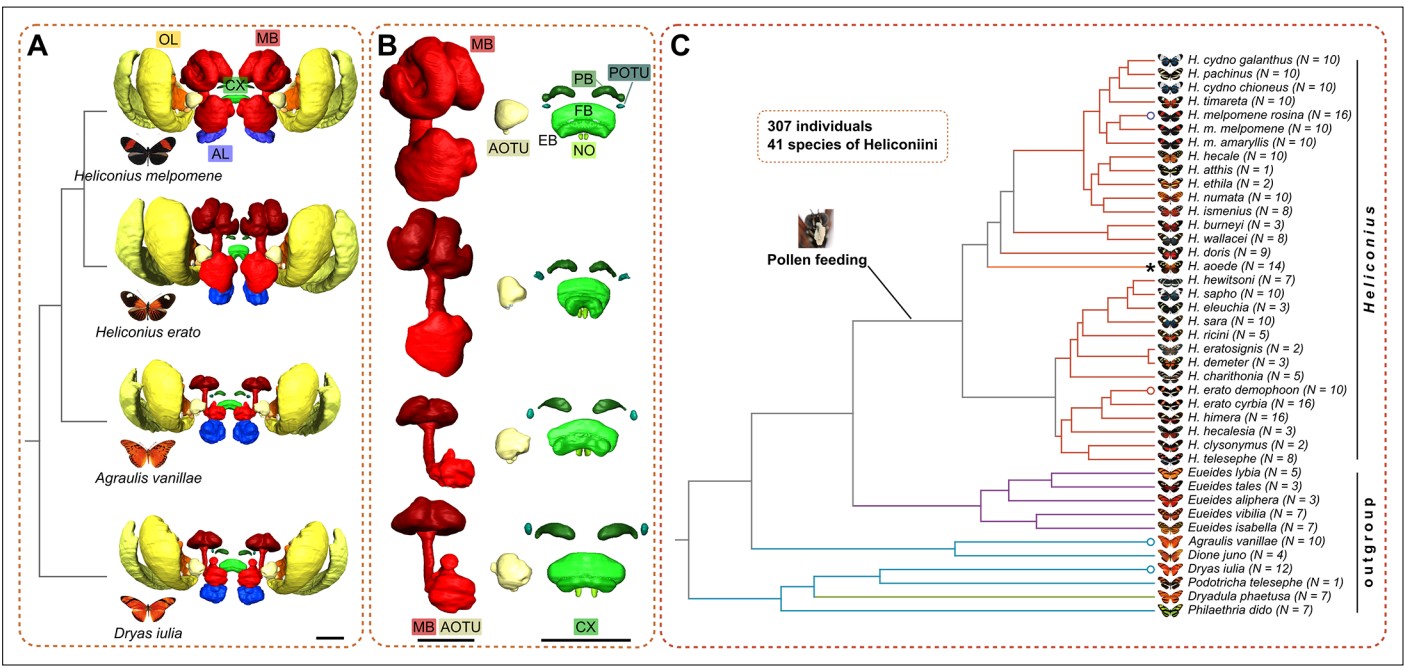

**Figure 1.** Introduction to the Heliconiini system used in this study. (**A**) The Heliconiini feature a previously reported expansion of the mushroom bodies in *Heliconius sp.* concomitant with a dietary innovation of pollen feeding. Shown are 3D segmentations of the OL, AL, MB, CX, AOTU and POTU in the brain of four exemplary species. (**B**) Examining the CX of these four species already indicates that CX volume seems conserved, relative to variation in the MB. Shown are separate 3D segmentations of the MB and AOTU as well as the CX and POTU, with the CX and POTU consistently enlarged (i.e. not to scale to the MB models). (**C**) To assess divergence in volumetric investment in these structures we used a dataset of 307 individuals of 41 species of Heliconiini. Depicted is the phylogeny (*Cicconardi et al., 2023*) with species names, appearance, and number of individuals per species. Species shown in A and B are indicated here by a circle at the end of each edge. Colour indicates focal groups here and elsewhere (*Couto et al., 2023*). The asterisk at the branch of *Heliconius aoede* indicates a secondary loss of pollen feeding. Scale bars are always 250 μm. Abbreviations: OL optic lobe, AL antennal lobe, MB mushroom bodies, CX central complex, AOTU anterior optic tubercle, POTU posterior optic tubercle, PB, protocerebral bridge; FB, fan-shaped body; EB, ellipsoid body; NO, noduli.

a subregion, called the fan-shaped body (FB), via mushroom body output neurons (MBONs; *Hulse et al., 2021*). This likely allows insects to flexibly modify their navigational goal based on internal preferences, experience-related olfactory and visual cues, and behavioural context (*Matheson et al., 2022*; *Sun et al., 2021*). Such behavioural flexibility is of high importance during spatial foraging where the same location is repeatedly visited over time to exploit a reliable resource, while incorporating behavioural responses to new opportunities or threats (*Ohashi and Thomson, 2009*; *Lihoreau et al., 2013*).

Interestingly, while the mushroom bodies repeatedly show patterns of adaptability through volumetric expansion and internal structural change across insects (*Kuwabara et al., 2023*; *Farris, 2013*; *Farnworth et al., 2024a*; *Couto et al., 2023*; *Farris and Schulmeister, 2011*), the size and general morphology of the central complex appears to be more conserved, potentially reflecting evolutionary constraints (*Figure 1A–B*; *Honkanen et al., 2019*). For example, an elaboration of spatial foraging and associated long-term memory is concomitant with an expansion of their mushroom bodies in some lepidopterans, as well as in Hymenoptera (*Farris, 2013*; *Couto et al., 2023*; *Farris and Schulmeister, 2011*; *Young and Montgomery, 2020*), but changes in the central complex associated with spatially faithful foraging are less well characterised, and apparently less prominent (*Sayre et al., 2021*). However, their mutual functions in spatial foraging behaviour predict that adaptive changes in one of these neural circuits could have the potential to influence the structure and function of the other, resulting in a pattern of coevolution. As such, the contrasting degrees of interspecific variation and functional interdependence provide an intriguing basis to make inroads into the evolutionary dynamics underlying neural circuit changes across the anatomical boundaries of brain components.

Here, we provide a closer examination of whether changes in mushroom body architecture co-occur with altered central complex investment or structure. We leverage a closely related tribe of Neotropical butterflies, the Heliconiini, as a model system. Within Heliconiini, the genus *Heliconius* exhibits a pattern of spatially faithful foraging behaviour, consistent with trap-lining, to efficiently forage pollen from spatially distributed, but temporally reliable, floral resources (*Young and Montgomery, 2020*; *Moura et al., 2022*; *Moura et al., 2024*; *Young et al., 2024*; *Hodge et al., 2025*). To form spatially faithful foraging routes, cues associated with these sites need to be recalled based on time-stable, long-term memory of spatial locations (*Doussot et al., 2024*; *Collett and Collett, 2002*; *Collett et al., 2013*). This complex foraging behaviour is likely supported by changes in the circuits involved in learning, memory, and spatial navigation, with *Heliconius* exhibiting the capacity to learn spatial information at large distances (*Moura et al., 2023*), likely using landscape features (*Moura et al., 2024*), and an enhanced capacity for learning complex visual cues (*Couto et al., 2023*) and to retain visual memories for prolonged periods of time (*Young et al., 2024*; *Hodge et al., 2025*). Interestingly, other genera within the Heliconiini share life history traits, habitats, and wider ecologies with species of *Heliconius* (*Brown, 1981*; *Hebberecht et al., 2022*). However, they do not use pollen as a food source and are not known to form long-term stable home ranges or perform the derived foraging strategies of pollen feeding *Heliconius* (*Young and Montgomery, 2020*). This increased reliance on spatial memory for faithful patterns of foraging in *Heliconius* suggests that divergent selection pressures between *Heliconius* and their closest relatives will have acted on the navigation circuits of the brain. Indeed, this stark behavioural divergence of a pollen feeding innovation coincided with a massive expansion of mushroom body volume, and structural changes in both the input sites, reflecting specialisation toward visual processing, and the output sites, reflecting possible specialisations in specific memory types (*Farnworth et al., 2024a*; *Couto et al., 2023*; *Young et al., 2024*). These adaptations in the mushroom body occur without apparent changes in upstream sensory pathways (*Couto et al., 2023*; *Hodge et al., 2026*), but the effects on downstream circuits have not been examined yet. The relatively shallow phylogenetic distances across Heliconiini, and their similarity in most behavioural domains provide a unique context in which to disentangle the effects of mushroom body expansion on interacting circuitries, including the central complex. Indeed, these circumstances permit us to test the hypotheses that modifications in the mushroom bodies either occurred in isolation from other integrative centres, or that they occurred in concert with specific changes in centres, such as the central complex. This provides insights into the functional flexibility of two interacting, integrative centres across evolutionary time.

To understand how the cognitive demands of pollen feeding and systematic foraging shaped the evolution of the central complex in *Heliconius*, volumetrically as partner of the mushroom bodies, but

also in terms of its finer anatomy, we combined phylogenetically broad sampling of neuropil volume with fine anatomical descriptions among representatives of key lineages. To assess evidence of volumetric changes in the central complex and associated neuropils, we drew data from a large dataset of immunostained brains from 307 individuals of 41 species, to determine patterns of evolution using phylogenetic modelling. We then used a series of stainings, primarily targeting classic neurotransmitters, to examine fine anatomical patterns beyond volumetric effects in a targeted group of species and individuals. The combination of phylogenetically deep data and fine-anatomical approaches in select species allowed us to examine patterns of mushroom body and central complex co-evolution in the adaptation to a novel food source.

## Results

We examined the evolution of central complex (CX), anterior optic tubercle (AOTU), and posterior optic tubercle (POTU) investment, in the context of the previously characterised expansion of the mushroom bodies (MBs) across Heliconiini. We used a combination of phylogenetic comparative analysis across a large dataset of brains immunostained against the structural marker synapsin in 41 species and 307 individuals, and more targeted sampling of species that represent the behavioural and neuroanatomical diversity of Heliconiini for more fine-scale assessments of patterns of divergence in substructures of the CX with various antibodies (*Figure 1A–B*). In doing so, we provide the first comparative assessment of the CX in this emerging case study in neural evolution. The CX spans the midline and consists of four neuropils, the protocerebral bridge (PB), the noduli (NO) as well as the fan-shaped body (FB) and ellipsoid body (EB) which together constitute the central body (CB; *Figure 1B*). The AOTU is localised antero-lateral to the CX and is a key processing centre of the insect compass pathway of the central brain, transmitting information from the AOTU via tubercular-bulbar (TuBu) neurons to the bulb (BU) where ellipsoid body ring (ER) neurons then transfer the information to columnar neurons of the CX (*Kandimalla et al., 2023*; *Homberg et al., 2023a*; *Heinze and Reppert, 2012*; *Figure 1B*). The POTU is an optic glomerulus of the central brain that lies postero-lateral to the protocerebral bridge, close to the posterior optic commissure. The POTU receives synaptic input from the accessory medulla of the optic lobe and forms tight connections with the protocerebral bridge through polarisation-sensitive neurons in some insect species (*Beetz et al., 2015*; *El Jundi and Homberg, 2010*; *Heinze and Homberg, 2007*). Both the AOTU and POTU are easily discernible neuropils and form clear connections with different parts of the CX, allowing us to test for effects of circuit change in tightly connected sister circuits beyond the CX.

### Isolated impact of mushroom body expansion within the central complex, AOTU, and POTU circuit

To assess whether cognitive traits associated with pollen feeding not only evolved concurrently with expansions of the mushroom bodies (MB) (*Couto et al., 2023*) but also with volumetric changes in the CX and its closely associated visually linked neuropils, the AOTU and POTU, we determined volumes of all CX neuropils (PB, FB, EB, and NO), as well as the combined volume of the three subunits of the AOTU, and the POTU, in our phylogenetic dataset of 41 species of Heliconiini butterflies (*Figure 1C*).

To initially determine whether sex should be merely a control variable in subsequent testing or whether there were effects examinable for closer inspection, we started with a logical nested model design first (model a with only sex as test variable, b with sex and genus effects, c with their interaction), generated MCMCglmm models, and compared the DIC. Using the simplest model with the lowest DIC, we used this to examine sex differences. We found that the simplest model [total CX/AOTU/POTU ~rCBR + sex+(1|Phylogeny)] was performing very close to others, hence used this to make inferences. There were significant sex differences in the CX and the AOTU and none in the POTU (whole CX: $P_{MCMC}$ = 0.026; AOTU: $P_{MCMC}$ = 0.004; POTU: $P_{MCMC}$ = 0.737), but these sex differences were very small indeed in absolute terms (*Figure 2—figure supplement 1*). In absence of any obvious biological inferences we would be able to place on this variation, we continued using sex as a control variable, as to account for variance through sex only.

We first examined the effects of mushroom body expansion on CX evolution by testing the relationships between the presence of pollen feeding (PF), clade membership (defined as groups with distinct degrees of expansion, identified previously *Couto et al., 2023*) and mushroom body size

(*MB \* PF; MB \* Clade*) on the volumes of the CX, its constituent neuropils as well as the AOTU and POTU (*Supplementary file 1* for model overview). Throughout the CX, AOTU, and POTU, as well as within the CX neuropils, no significant changes in volumetric investment were detected that were associated with pollen feeding, clade membership, and mushroom body size (*Figure 2A, B*, *Figure 2—figure supplement 2*, *Supplementary file 1*). We observed a very tight association with our allometric control (central brain size minus focal neuropils; rCBR). We did identify significant associations between MB size on CX size (Indications in *Figure 2A–B*), isolated from the presence of pollen feeding or clade membership. However, this reflects the remaining effects of allometric scaling between brain components, rather than non-allometric shifts reflecting the effects of divergent selection.

We next wanted to corroborate the distinct absence of volumetric expansions in the CX, coincident with mushroom body expansion or otherwise, by examining evolutionary rates of change in CX, AOTU, and POTU volumes in comparison with rates of MB expansion (*Couto et al., 2023*). Previously, (*Couto et al., 2023*) identified very high evolutionary rates of mushroom body expansion at the base of *Heliconius* where pollen feeding evolved. In contrast, we found a distinct absence of any changes in the evolutionary rate of CX size (*Figure 2C/D*, *Figure 2—figure supplement 3B*). Analogous analyses for the AOTU and POTU revealed similar levels of consistent, low rates of evolution during key evolutionary periods of ecological transition (*Figure 2—figure supplement 3B–C*). Minor evolutionary rate changes were observed in select terminal branches (e.g. for the CX in *Heliconius burneyi, hecale* or *Eueides aliphera; Figure 2D*, *Figure 2—figure supplement 3B*) but are not associated with known behavioural changes or MB expansion (*Figure 2C*). We therefore conclude that the neural changes accommodating cognitive demands for pollen feeding did not include volumetric expansion or increased investment in the CX (and its neuropils), or in the wider visual pathway that feeds into the CX (AOTU and POTU). The CX therefore evolves as a distinct system compared to the major volumetric expansion in the MB.

## Mosaic patterns of co-dependency between central complex neuropils, AOTU, and POTU

We next explored how the different CX neuropil and AOTU/POTU volumes co-vary with each other, and to what extent these interspecific scaling relationships of volumes mirrored known functional links between neuropils. Moreover, we tested whether these co-evolutionary relationships were altered during mushroom body expansion.

We first examined whether variation in each structure is determined by variation in all other structures using multiple regression analyses. This revealed a mosaic pattern of co-variation, where the AOTU positively scaled with FB and EB, the POTU with PB, the FB with EB and the EB with the PB, but all other structural pairings scaled independently of each other (*Figure 3A*). To determine whether any of these dependencies were impacted by the origin of pollen feeding we re-ran the significant pair-wise regressions including presence of pollen feeding as a factor. This revealed that all scaling relationships were robust to the presence or absence of pollen feeding, indicating conserved co-evolutionary relationships across the tribe.

Indeed, we found that the presence of dependencies can illustrate, in most cases, anatomical and functional links that may mutually constrain variation in these structures through characteristic cell types (*Figure 3B and C*), whose nomenclature derives from the neuropils they connect. The most relevant for the scaling relationships we observed are the EPG (connecting the **E**B, **P**B, and part of the lateral complex, the **G**all), PFN (PB-FB-NO), PEN (PB-EB-NO), PFL (PB-FB-LAL), ER (BU-EB), and Δ7 (POTU-PB) neurons (*Hulse et al., 2021*; *Kandimalla et al., 2023*; *Homberg et al., 2023a*; *von Hadeln et al., 2020*; *Jahn et al., 2023*; *Heinze and Homberg, 2008*; *Pfeiffer, 2023*; *Heinze, 2024*; *Figure 3B*). The majority of results from pairwise scaling comparisons match what we would expect from these known cell types and prominent connectivity patterns, but some do not. Although these results require further investigation, we interpret our results as evidence that some of the within-scaling co-evolutionary relationships of the CX may be determined by functional co-dependence, but that the evolution of pollen feeding and mushroom body expansion has not altered these internal relationships. This would be consistent with a largely conserved circuit architecture.

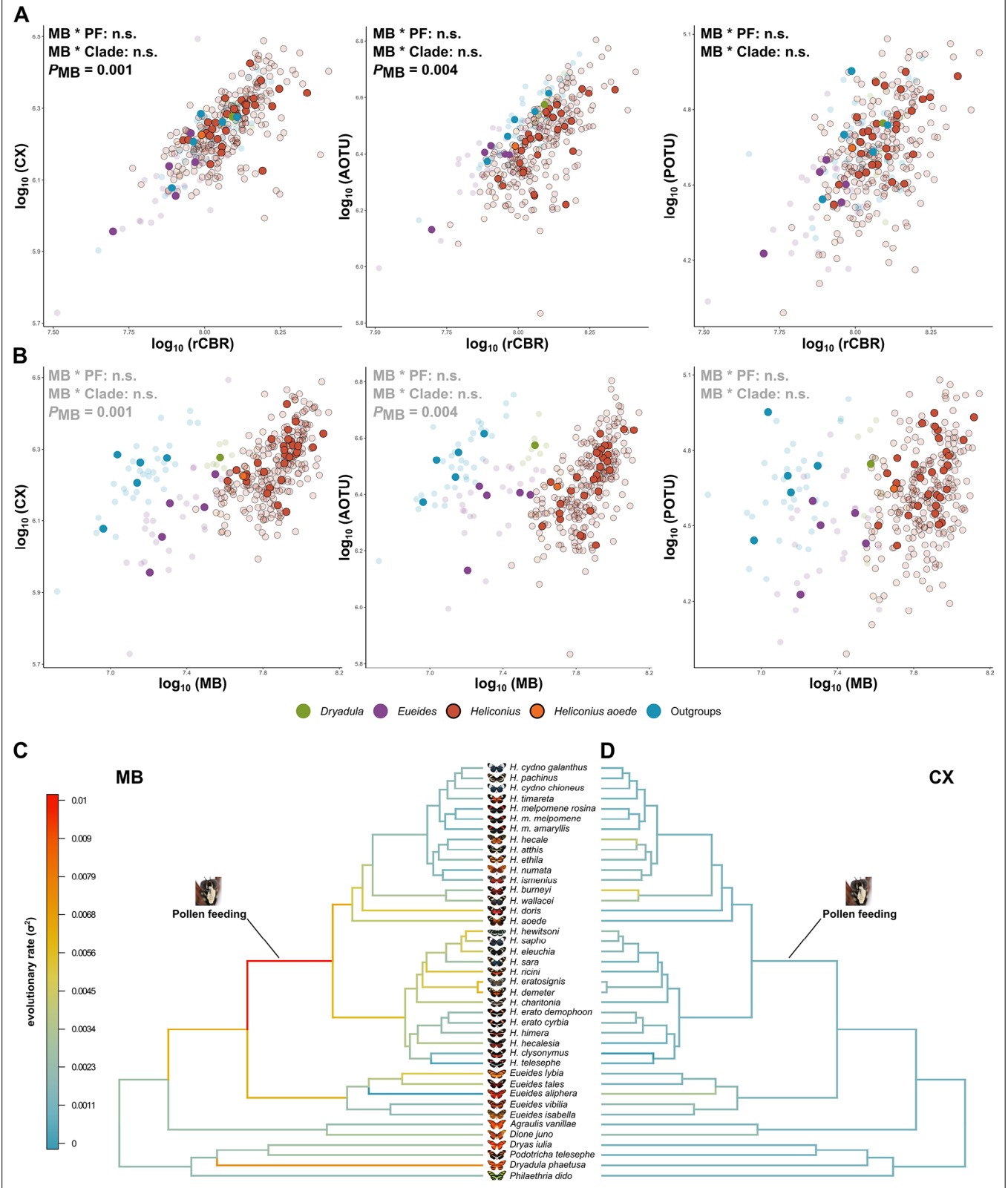

**Figure 2.** The central complex, the AOTU and POTU do not show pollen-feeding linked patterns of volumetric expansion or increased evolutionary rates across a data set of 307 individuals and 41 species of Heliconiini butterflies. (**A/B**) Assessments of the effects of mushroom body size interacting with pollen feeding and with previously identified expansion events in the sum of all CX neuropils (total CX), AOTU, and POTU, and in separate CX neuropils (see *Figure 2—figure supplement 2*). Shown are each neuropil against a measure of whole brain size, rest of the central brain (rCBR; **A**), as

*Figure 2 continued on next page*

*Figure 2 continued*

well as against MB size (**B**). Results are indicated in each panel; first, the effects of each interaction by assessment of DIC differences, and then the significance of the relationship between the MB and each neuropil. Colour coding was done according to clade differences in mushroom body size identified previously (***Couto et al., 2023***). Solid data points indicate species averages, while opaque circles indicate individual data points. Black contours indicate *Heliconius sp.* data points. (**C/D**) Analyses of evolutionary rates of MB size relative to rCBR (**C**), and CX size relative to rCBR (**D**), which reveals two very distinct evolutionary patterns. In the MB (**C**), particularly high rates of evolution have previously been identified on the branch leading to *Heliconius* (***Couto et al., 2023***), coincident with the innovation of pollen feeding. At the same branching point, we see relatively low rates of evolution in CX size (**D**). Both phylogenies are shown at the same scale (see *Figure 2—figure supplement 3* for a CX tree with absolute scale of evolutionary rates). The pollen feeding photograph is kindly provided by Sebastian Mena.CX central complex, AOTU anterior optic tubercle, POTU posterior optic tubercle, MB mushroom bodies, rCBR rest of the central brain. PF pollen feeding.

The online version of this article includes the following figure supplement(s) for figure 2:

**Figure supplement 1.** Differences of sex identity onto the relative size of the CX, AOTU and POTU across a data set of 307 individuals and 41 species of Heliconiini butterflies.

**Figure supplement 2.** Central complex neuropils do not show pollen-feeding linked patterns of volumetric expansion across a data set of 307 individuals and 41 species of Heliconiini butterflies.

**Figure supplement 3.** Evolutionary rate analysis in relative CX (**A**), AOTU (**B**) and POTU (**C**) size evolution corroborate an isolated increase of evolutionary rate in the MB concomitant to pollen feeding but not in areas shown here.

## Conserved tract and synaptic architecture in the Heliconiini central complex

We next sought to test our interpretation more closely by assessing whether other factors beyond size are obscured by the conserved volume of the CX. We first examined the CX with general labels to examine the complete CX architecture of a representative outgroup, *Dryas iulia*, and a pollen feeding *Heliconius*, *Heliconius erato* (***Figure 4***). We then used several antibodies targeting neurotransmitters and neuropeptides, as well as bulk injections of portions of cell populations labelled, to more closely

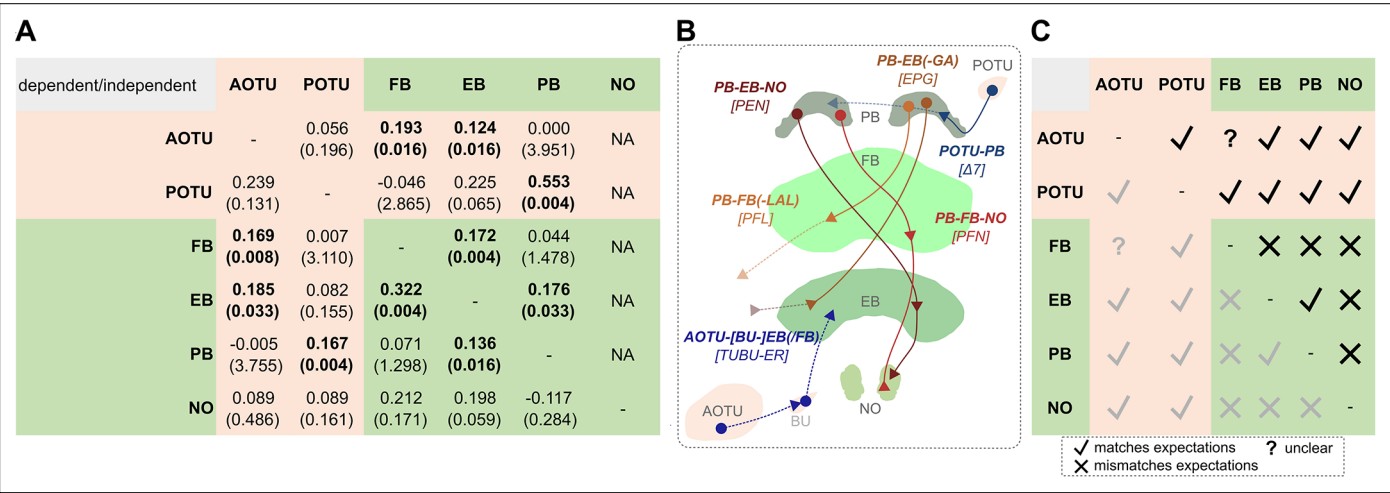

**Figure 3.** Central complex neuropils and associated areas show different patterns of scaling with each other. (**A**) Tests of significant scaling relationships between each substructure on the left, as dependent variable, and all others as independent variables. Each cell indicates the test statistic (posterior mean) and the p value in parentheses, and if significant is shown in bold. As values for NO were only available for parts of the total dataset, noduli were not included as independent variables. (**B**) Prominent cell types that interconnect central complex neuropils and the AOTU and POTU, which may potentially explain patterns of neuropil scaling, if positive scaling relationships indicate co-evolution among functionally interdependent structures. Cell type depictions are examples with localisation inside each neuropil being purely visual (as well as their colour), while triangles indicate approximate output sites. Namings first refer to the connected neuropils, then in brackets to established nomenclature, based on current literature (***Hulse et al., 2021***; ***Kandimalla et al., 2023***; ***Homberg et al., 2023a***; ***von Hadeln et al., 2020***; ***Jahn et al., 2023***; ***Heinze and Homberg, 2008***; ***Pfeiffer, 2023***; ***Heinze, 2024***). (**C**) A depiction of where our data showing significant scaling relationships (**A**) matches (check mark), or mismatches (**X**), expectations that are based on prominent (mostly columnar) neuron classes and connections (**B**). Where expectations are unclear, we have annotated the comparison with (?). Specifically, we would normally not expect positive scaling between AOTU and FB, but this may be explained by findings in ***Figure 7*** and an increased population of ER neurons projecting to the FB. Generally, this diverse pattern indicates that more variation occurs throughout the system than can be captured in this volumes-based analysis. AOTU anterior optic tubercle, POTU posterior optic tubercle, PB, protocerebral bridge; FB fan-shaped body, EB, ellipsoid body; NO noduli, GA gall.

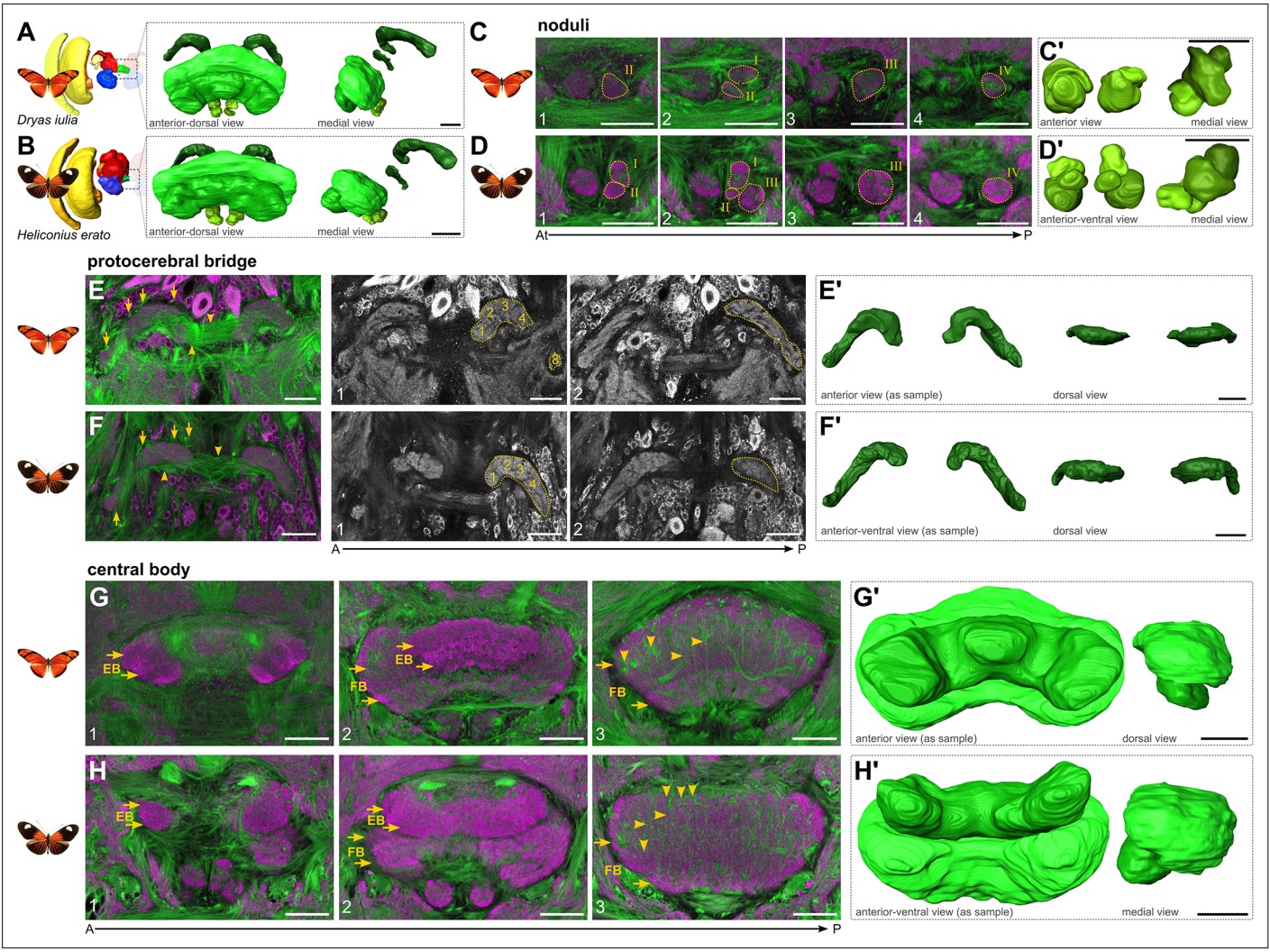

**Figure 4.** Tract and synaptic labelling reveal a conserved central complex architecture in Heliconiini butterflies. (**A/B**) Double labelling of tubulin (green) and horseradish peroxidase (HRP; magenta) was used to reveal the synapse-dense areas of the central complex as well as its underlying tract systems. (**C/D**) Noduli architecture reveals, tentatively, a conserved pattern of four main domains, but with very pronounced asymmetry between hemispheres. (**E/F**) Tract bundles close to the PB associate with DM1-4/WXYZ tracts (arrows). Tangential neuron bundles project across hemispheres through the PB (arrowheads). HRP alone (grey) allows a rough approximation of vertical columns in the PB (**a–h**). (**G/H**) Labelling reveals two distinguishable layers in the fan-shaped body while additional staining elsewhere reveals further detail (arrows in G/H-2/3). Thicker tract conflations indicate the columnar architecture determined through the four columnar neuron bundles (arrowheads in G/H-3). Labelling in the EB reveals two pronounced layers (arrows in G/H-1/2), while obvious columns could not be indicated. PB protocerebral bridge, FB, fan-shaped body; EB ellipsoid body. A, anterior; P, posterior. Scale bars are 50 μm. See animated 3D segmentations in *Figure 4—video 1; Figure 4—video 2*, and annotated stacks in file repository.

The online version of this article includes the following video(s) for figure 4:

**Figure 4—video 1.** Animated 3D segmentations of *Dryas iulia* central complex.

https://elifesciences.org/articles/107589/figures#fig4video1

**Figure 4—video 2.** Animated 3D segmentations of *Heliconius erato* central complex.

https://elifesciences.org/articles/107589/figures#fig4video2

examine the anatomy of the CX (*Figures 5 and 6*, *Figure 7—figure supplement 1*), and to identify anatomical divergences in selected markers (*Figures 7 and 8*, *Figure 7—figure supplement 2*).

We first used general antibodies targeting acetylated tubulin to visualise neuronal tract anatomy, and horseradish peroxidase (HRP) to determine synaptic areas (*Jan and Jan, 1982*). Not surprisingly, these general labels revealed a very conserved architecture and a stereotypical makeup of the CX, very comparable to the Monarch butterfly CX (*Figure 4A/B*; *Heinze et al., 2013*), briefly summarised as:

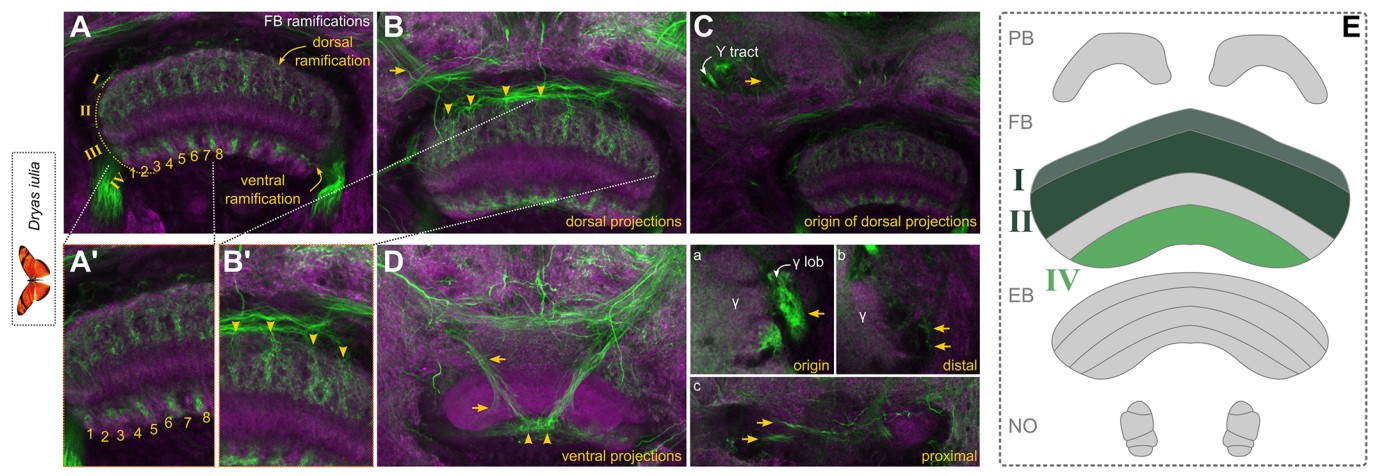

**Figure 5.** Mass injections in *Dryas iulia* reveal columnar architecture and input into the fan-shaped body (FB). (**A**) A single injection into the superior median and lateral protocerebrum (SMP/SLP) of *Dryas iulia* has revealed prominent labelling of the fan-shaped body, particularly layers I, II (dorsal projection), and IV (ventral projection). (**B**) Input pathways into these layers from a dorsal set of projections through four input tracts into the fan-shaped body (arrowheads), originating from the SMP/SLP region (**C**, arrow), whereas A' and B' are close-ups of A and B, highlighting the unique columnar architecture of the fan-shaped body revealed through this labelling. (**D**) Ventral projections also seem to originate from the SMP/SLP region but are hard to trace. (**a–c**) depict close-ups of an interesting input pathway from γ lobe and lobelet into the fan-shaped body through the ventral projection. (**E**) Schematic of labelling in generic central complex. Abbreviations: FB fan-shaped body, lob, lobelet of the mushroom body. Scale bars equal 100 μm.

A. Noduli: We tentatively defined four layers of the noduli (*Figure 4C/D*). However, the exact boundaries were difficult to determine without additional markers as any tracts, which often can be used as suitable landmarks for compartment borders (*Farnworth et al., 2022*), were too small inside the noduli. We also note a general variability in left and right-side symmetry of the noduli. In part, this matches previous observations into nodulus anatomical variation (*Heinze, 2024*).

B. Protocerebral bridge: The protocerebral bridge was generally conserved (*Figure 4E/F*). Like in other lepidopteran species (*Heinze et al., 2013*), the protocerebral bridge is split into two halves that are connected across the brain hemisphere via a commissural tract. Tubulin allowed us to determine tract conflations that form the WXYZ tracts (also known as DM1-4 tracts) that interconnect the protocerebral bridge with the central body via columnar neurons (arrows in *Figure 4E/F*; *Farnworth et al., 2020*). We labelled a tentative columnar array of eight columns of the protocerebral bridge in each hemisphere (*Figure 4E/F–1*).

C. Fan-shaped body: In the fan-shaped body, we were able to distinguish a dorsal and a ventral layer based on the tubulin and HRP stainings. Note that, with subsequent neurotransmitter stainings and mass injections, we were able to further subdivide the fan-shaped body into a total of four layers (arrows in *Figure 4G/H-2/3* and *Figures 5 and 6*). Tracts enter the synapse-dense neuropil in very stereotypical fashion, indicative of the tracts that correspond to the columnar neurons of the DM1-4 lineages, which give rise to the columnar architecture of the fan-shaped body (arrowheads in *Figure 4G/H–3*).

D. Ellipsoid body: In the ellipsoid body, we also observed two prominent layers through HRP staining (*Figure 4G/H-1/2*), but using subsequent stainings identified four layers (*Figure 6*). Note that the 3D segmentations in *Figure 4G' and H'* might suggest different shapes of the ellipsoid body, but this is rather due to slightly different positioning of the brain relative to the axis of imaging.

The layering of the fan-shaped body and its columnar architecture indicated by tubulin and HRP combinations was corroborated by mass injections of Dextran into the superior medial and lateral protocerebrum (SMP, SLP) of *Dryas iulia*, which revealed a distinct labelling of fan-shaped body layers I, II, and IV (*Figure 5A*), originating from dorsal as well as ventral projections (*Figure 5B–D*). With these injections we were able to determine fan-shaped body columns in layer IV particularly, and four tract conflations of the dorsal projections (arrowheads in *Figure 5B*), originating from the superior

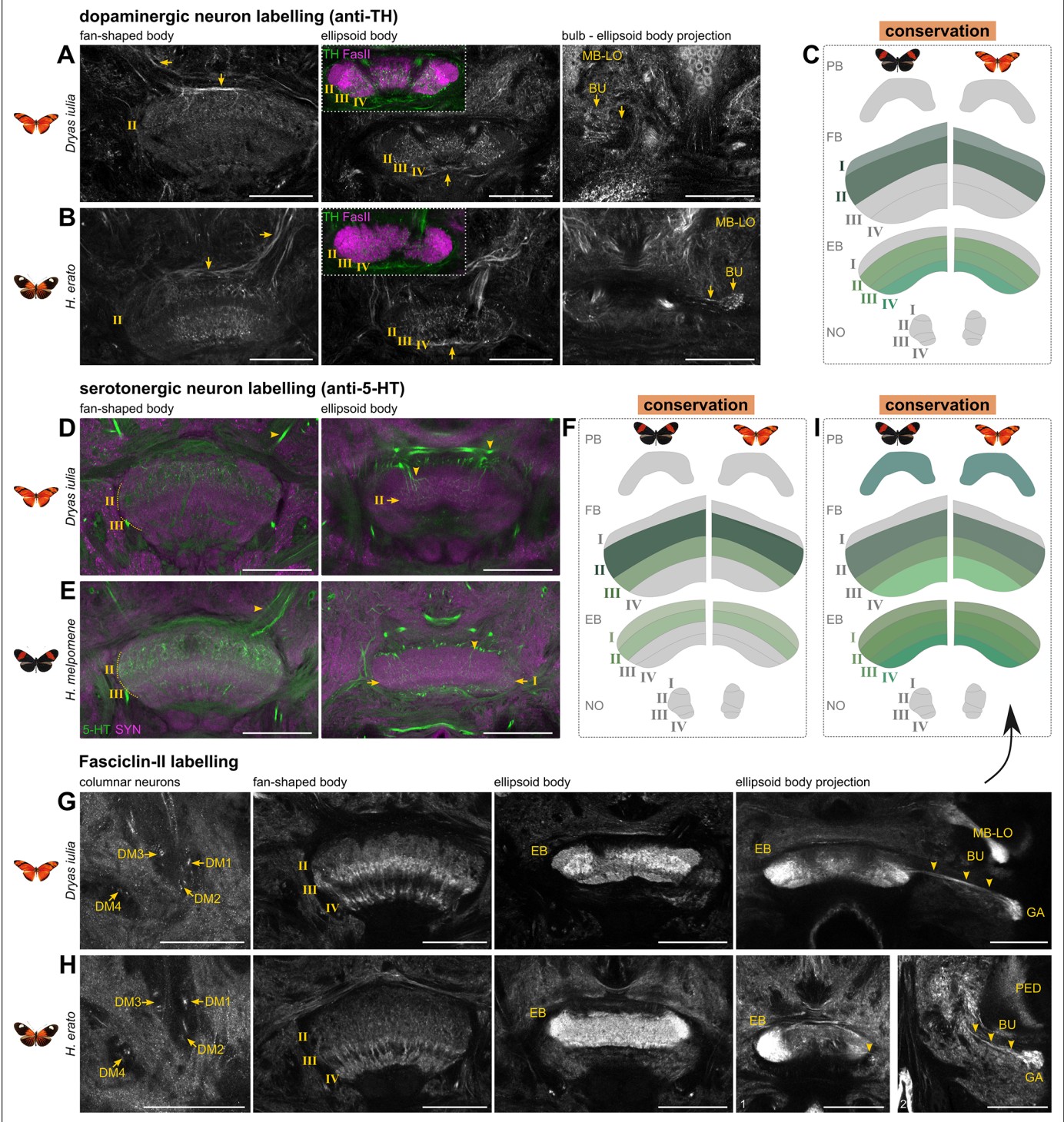

**Figure 6.** Comparative anatomical analysis of the Heliconiini central complex examining dopaminergic and serotonergic neurons and Fasciclin-II expression supports a largely conserved neurotransmitter expression and anatomy. Shown are original data and a schematic of the central complex summarising expression patterns inside the synaptic region of the central complex. (**A–C**) Patterns of Tyrosine Hydroxylase (TH) labelling of dopaminergic neurons depict weak labelling of fan-shaped body layers I and II, spot-like layering of ellipsoid body layers I-III, as well as projection pathways from the superior medial protocerebrum into the fan-shaped body (arrows), ventral projections into the ellipsoid body (arrows), and projections from the bulb into the ellipsoid body (arrows). (**D–F**) Patterns of 5-hydroxytryptamine (5-HT) labelling of serotonergic neurons depict prominent labelling into layers II and III of the fan-shaped body, from projections in the superior medial protocerebrum (arrowheads), as well as showing spot-like labelling of the ellipsoid body layers I and II from ventral projections as well as dorsal projections (arrowheads). (**G–I**) Fasciclin-II labelling of a subpopulation of columnar neurons across all lineages DM1-4 emerging into prominent labelling of layers II-IV of the fan-shaped body and all layers

*Figure 6 continued on next page*

Figure 6 continued

in the ellipsoid body. In addition, projections into the gall from the ellipsoid body are shown (arrowheads). Abbreviations: PB protocerebral bridge, FB fan-shaped body, EB ellipsoid body, NO, noduli; MB-LO, mushroom body lobes; BU bulb; DM1-4, dorso-medial lineage 1–4; GA, gall. Scale bars equal 100 µm.

medial protocerebrum (*Figure 5C*). Some of the ventral projections seemed to directly originate from the γ lobe, a portion of the mushroom body, thus potentially labelling projections of mushroom body output neurons into the fan-shaped body (*Figure 5a–c*; *Li et al., 2020*; *Hulse et al., 2021*).

Overall, the injections, standard labels and neuromodulatory stainings revealed two major groups of input channels consistently across species. A ventral projection that enters from ventrolaterally, in most cases into the central body (*Figures 5D and 6A-F*), and one that originates from the superior lateral or medial protocerebrum, often with its dendritic fields unclear, and projects dorso-laterally or medially into the central body (*Figures 5B-C and 6A-F*). These seem to be part of conserved supply channels, as they are also observed in *Drosophila* connectome data and data of the hemimetabolous *Schistocerca sp.* (see Figures 5, 6 and 8 in *Kandimalla et al., 2023* and information in *von Hadeln et al., 2020*) with recognizable projection patterns, despite strong shape differences.

## Conserved patterns of serotonergic and dopaminergic neurotransmitter expression in central body layers

To more closely assess fine anatomical details of the Heliconiini CX and identify potential species differences, we performed neurotransmitter (GABA, Dopamine, Serotonin; *Figures 6 and 7*, *Figure 7—figure supplement 1*), neuropeptide (Allatostatin A, *Figure 8*), and cell adhesion molecule (Fasciclin-II) stainings (*Figure 6*). This group of stainings served three overlapping purposes. First, to reveal closer details of general anatomy of the Heliconiini CX, with particular emphasis on central body layering, which proved more difficult with general markers (*Figures 4 and 6C/F/I/L*). Second, to examine differences in the expression of important neurotransmitters. And third, through more sparse labelling, to identify any anatomical and structural differences that might be obscured by volume as well as general markers.

We first identified conserved patterns in dopaminergic and serotonergic neurons as well as in neurons labelled by Fasciclin-II (*Figure 6*). To label dopaminergic neurons, we used a Tyrosine Hydroxylase (TH) antibody, which resulted in much less prominent staining inside the CX itself (*Figure 6A–C*). However, through dorsal projections above the fan-shaped body, we could detect innervations of fan-shaped body layer I and II. Layer I was only very faintly labelled, with arborisations leading back to cell bodies near the mushroom body calyx. These neurons might be homologs to neurons reported in the moth *Manduca sexta* (*Timm et al., 2021*). We detected spot-like labelling of ellipsoid body layers II-IV, which are likely generated by neurons that project from ventral and dorsal locations that might include ER neurons with inputs in the bulb (*Figure 6A–C*), and/or neurons homologous to EXR2/TL5 neurons (*von Hadeln et al., 2020*). Within the constraints of the information provided in these stainings, the pattern of dopaminergic staining was highly conserved across species (*Figure 6C*).

To identify serotonergic neurons we used a 5-hydroxytryptamine (5-HT) antibody. Like in other insects (*Homberg et al., 2023b*), 5-HT staining revealed a prominent labelling of fan-shaped body layer II and weaker labelling of layer III through a small number of neurons in Heliconiini (*Figure 6D–E*). The cell bodies of these neurons are situated at the most posterior end of the brain, surrounding the protocerebral bridge and anterior sections of the calyx. Beyond the branches in the CX, these neurons seem to arborize inside the SLP and SMP. Layer II of the ellipsoid body was faintly labelled in both species. Particularly, the pattern of strong layer II labelling was well conserved across other Heliconiini species (*Figure 7—figure supplement 1C–D*). Taken together, we did not observe any pronounced differences of serotonergic neuron expression across *Heliconius* and their non-pollen feeding outgroups (*Figure 6F*).

We also stained against Fasciclin II (Fas II), a cell adhesion molecule with broad neurodevelopmental roles that was previously used to reveal more specific detail in insect mushroom bodies (*Farnworth et al., 2024a*; *Crittenden et al., 1998*), but that also showed interesting patterning in the CX. In contrast to the neurotransmitter stainings, this staining labelled a portion of columnar neurons of each group of DM1-4 lineages, that is a subset of the WXYZ tracts (*Figure 6G–I*) which resulted in prominent labelling of layer III and IV of the fan-shaped body as well as weaker labelling of layer

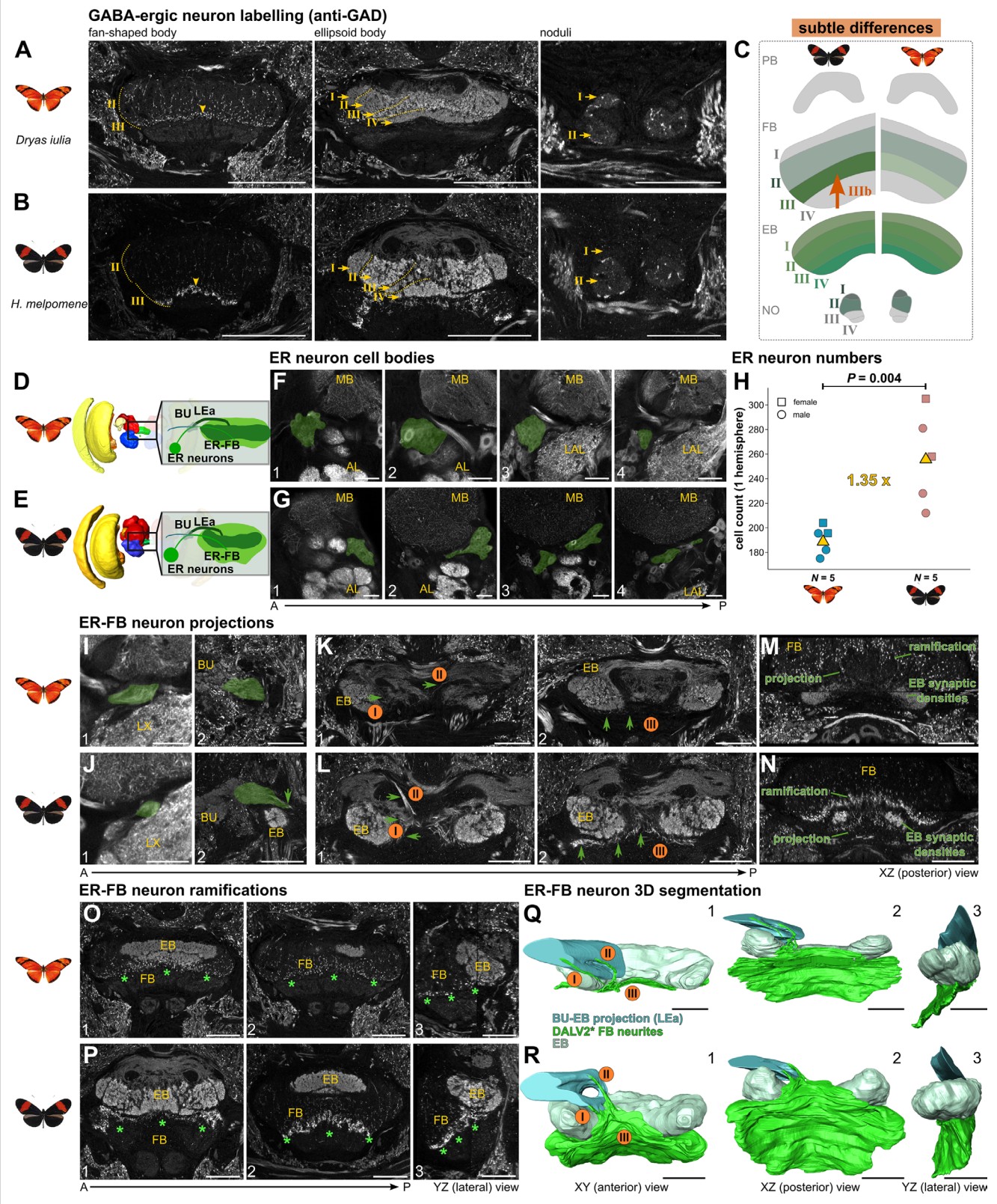

**Figure 7.** Anatomical and statistical analysis of GABA-ergic DALv2 lineage-derived ER neurons showing higher innervation of the FB in pollen feeding *Heliconius melpomene* versus *Dryas iulia*. (**A–C**) Patterns of GAD (glutamate decarboxylase) labelling of GABA-ergic neurons depicting spot-like labelling in the fan-shaped body layers II and III, complete labelling of the ellipsoid body and spot-like labelling of nodulus layers I and II. (**D/E**) Location of the GABA-ergic cell group and its projections in focus relative to the brain. (**F/G**) Position of cell bodies, nestled between the mushroom body lobes

*Figure 7 continued on next page*

*Figure 7 continued*

and antennal lobe at the anterior surface of the brain, along four different positions of the anterior-posterior axis ('1–4'). (**H**) Quantification of cell body numbers in both species revealed a significant increase in *Heliconius*. (**I/J**) ER neurite close to the cell bodies at two different positions ('1–2'). (**K/L**) Projection of ER neurites into the central body, with a focus on the projections into the FB. 'I/II' indicate two tracts that project ventrally and form a fibre that sits ventrally to the EB ('III') (**M/N**) A perpendicular view onto the projections labelled with 'III' show the relative position of the beginning synaptic EB neuropil as well as the ER projections and ramifications leading into the FB. (**O/P**) Focus on the ramifications of the ER-FB neurons inside the FB at two different A-P axis points ('1–2'), as well as a lateral view ('3'). (**Q/R**) 3D segmentations of the ER neurites with focus on the FB contributing portion, with three different viewpoints ('1–3'). Light blue depicts the massive projections into the light-grey labelled EB, with green showing the projections and ramifications into the FB. All immunostaining depictions are based on an anti-GAD labelling. Abbreviations: BU, bulb; LEa, lateral ellipsoid body tract; ER ellipsoid body ring neurons, MB mushroom bodies, AL antennal lobe, LX, lateral complex; FB fan-shaped body, EB ellipsoid body.A, anterior; P, posterior. Scale bar equals 50 µm, except for the central body panels in A/B where it is 100 µm. See animated 3D segmentations in *Figure 7—video 1; Figure 7—video 2*, and annotated stacks in file repository.

The online version of this article includes the following video and figure supplement(s) for figure 7:

**Figure supplement 1.** GABA-ergic and serotonergic labelling in additional species.

**Figure supplement 2.** Segmentation of the bulb and its microglomeruli reveals conserved metrics despite increased ER neuron numbers in pollen feeders.

**Figure 7—video 1.** Animated 3D segmentations of *Dryas iulia* ring neuron innervation.

https://elifesciences.org/articles/107589/figures#fig7video1

**Figure 7—video 2.** Animated 3D segmentations of *Heliconius melpomene* ring neuron innervation.

https://elifesciences.org/articles/107589/figures#fig7video2

**Figure 7—video 3.** Animated 3D segmentations of *Dryas iulia* bulb.

https://elifesciences.org/articles/107589/figures#fig7video3

**Figure 7—video 4.** Animated 3D segmentations of *Heliconius melpomene* bulb.

https://elifesciences.org/articles/107589/figures#fig7video4

II. In addition, we also observed very prominent labelling in the entire ellipsoid body, revealing all four layers, particularly in *Dryas iulia* (*Figure 6G*). The staining further revealed a tract that connects the ellipsoid body to a small region of the lateral complex, called the gall (right most panels of *Figure 6G–H*; *Ito et al., 2014*). As this tract is typical for EPG neurons, this suggests that Fas II stained the compass neurons in the Heliconiini ellipsoid body (*Heinze et al., 2013*; *Wolff et al., 2015*; *El Jundi et al., 2019*). In summary, the observed patterns of these three stains reveal conserved details of anatomy across the Heliconiini.

## GABA-ergic ER neurons increase in number, with increased innervation of the fan-shaped body in pollen feeders

We next explored variation in GABA-ergic neurons and neurons that express Allatostatin A, which reveal evidence of deviation from the general conservation described thus far. GABA-ergic antibody staining using an anti-Glutamate decarboxylase (GAD) antibody resulted primarily in staining of ER neurons which we focus on chiefly (*Figure 7*, *Figure 7—figure supplement 1A-B*). Generally, however, this antibody labels conserved features (*Figure 7A–C*), specifically, we detected spot-like labelling in layers II and III of the fan-shaped body with projections from the SMP and SLP (matching expected pathways between mushroom bodies and CX and these regions *Hulse et al., 2021*), as well as layers I and II of the noduli with unclear, but anterior, origin.

Typical for GABA-ergic expression (*Homberg et al., 2018*), the most dominant and conserved labelling was in all four layers, I-IV, of the ellipsoid body (*Figure 7*). Anti-GAD staining reveals a pattern where large portions of neural lineages are labelled and the whole cells, including somata and branches, are evenly marked. This allowed us to quantify even densely packed cell groups typical for insect neurons, such as the DALv2 neural lineage and resulting ER neurons which we focussed on next. To identify this cell group, we cross-referenced our labelling with previously published work (*Homberg et al., 2018*). This revealed a characteristic and very prominent labelling of the ellipsoid body by a cell group whose cell bodies are positioned anterior-laterally to the ellipsoid body, nestled between the antennal lobe and mushroom body lobes (*Figure 7D–G*). This well-characterised group of cells is usually referred to as R/ER (ring) or TL2/3 neurons (tangential neurons of the ellipsoid body), while the lineage they are produced by is referred to as dorsoanterior lateral ventral 2 (DALv2; or

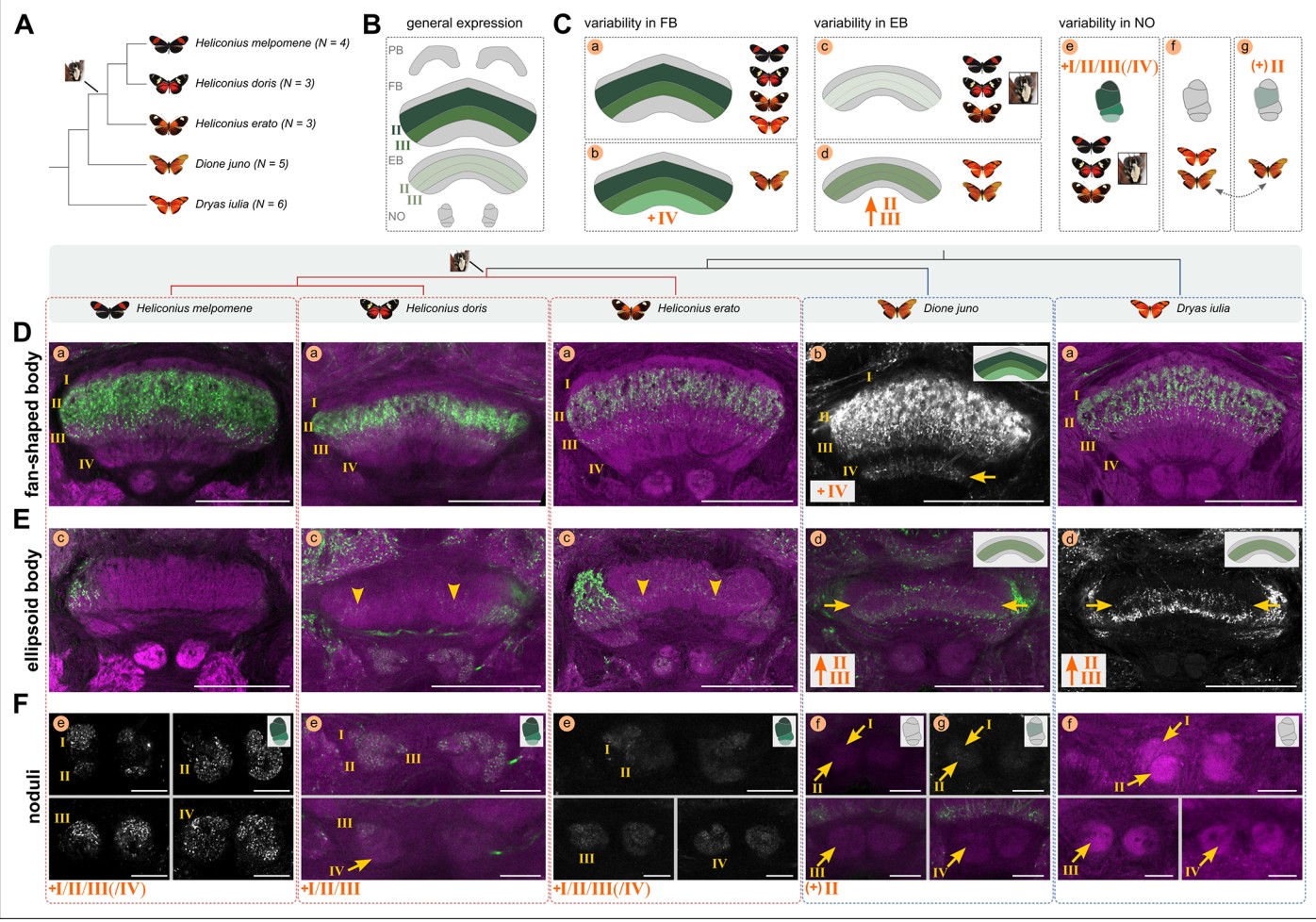

**Figure 8.** Anatomical analysis of Allatostatin A expression across five representative species of Heliconiini reveals variability consistent with behavioural innovations in *Heliconius*. (**A**) Summary of the dataset used to assess Allatostatin A variability, with the phylogeny of sampled species and indication of when pollen feeding evolved. (**B**) General expression pattern of Allatostatin A that is present throughout all species. (**C**) Schematic representation of the interspecific variation present in Allatostatin A expression in fan-shaped body, ellipsoid body, and noduli, with major differences indicated in orange lettering. If consistent with a shift in *Heliconius*, this was indicated by the pollen feeding photograph. Lower-case letters correspond to figure panels in **D-F**, where this variability of expression is depicted. Note that *Dione juno* varies in its noduli expression even between individuals, hence the double-headed arrow. (**D–F**) Variation in Allatostatin A expression (green or grey) across the five Heliconiini species. Synapsin was used as co-stain, depicted in magenta. Lower-case letters correspond to the schematic in C, with schematic depictions of the central complex neuropil in question indicated where consistent differences between species were identified. Arrowheads specifically point to difficult to identify expressions of Allatostatin A in the ellipsoid body of some species. Arrows point to differences described in the schematic and in text. Abbreviations: PB protocerebral bridge, FB fan-shaped body, EB ellipsoid body, NO noduli. Scale bars in D and E equal 100 µm and in F 25 µm.

anterior ellipsoid body; EBa1), and their tract is named either LEa (lateral ellipsoid fascicle, anterior part) or IT (isthmus tract; *Kandimalla et al., 2023*; *Homberg et al., 2018*; *Yang et al., 2013*). Here, we refer to these neurons, as well as those neurons projecting to the fan-shaped body (GU neurons in *Homberg et al., 2018*), as ER neurons due to their common developmental origin (*Kandimalla et al., 2023*; *Homberg et al., 2018*) and to simplify anatomical descriptions. The projection and ramifications inside the ellipsoid body were conserved across Heliconiini. Indeed these cells show strong conservation across insects (*Homberg et al., 2018*), as they are an essential part of the anterior visual (*Omoto et al., 2017*) and sky compass pathway (*Homberg et al., 2023a*), receiving visually guided signals from the AOTU and projecting those to the ellipsoid body (*Figure 7D–F*).

While we identified very strong conservation of projections to the ellipsoid body, we noticed a pronounced difference in a portion of projections leading into the fan-shaped body and a strong difference in signal inside layer III in our two focal species, *Heliconius melpomene* and *D. iulia*, as well as other representatives of the Heliconiini tribe (*Figure 7—figure supplement 1A–B*, *Figure 7*).

To understand how these differences could have occurred, we quantified ER neuron numbers in our focal species, and identified a significant difference, reflecting a 35% increase in *Heliconius* ($t=4.221$, $p=0.004$; *Figure 7H*). Most ER neurons will still produce the heavily conserved innervation of the ellipsoid body, but the additional neurons in *Heliconius* likely produce the increased ramifications we observed in layer III of the fan-shaped body.

To illustrate this connection, we next disentangled the projections that lead to the ellipsoid body from those that lead to the fan-shaped body. The whole cell group projects past the gall (GA) and lateral accessory lobe (LAL) to build a large conflation point constructing the bulb (BU) neuropil together with groups of neurons that project to it from the AOTU *Kandimalla et al., 2023*; *Homberg et al., 2023a* (*Figure 7I–J*). The projection, the LEa/IT, leads further into the central body where most neurons enter and terminate in the ellipsoid body (with a slightly modified shape of this projection between the two Heliconiini clades, *Figure 7Q/R–1*). In both species, we saw two major projections leading into the fan-shaped body, following a projection ventrally and posteriorly, away from the ellipsoid body (*Figure 7K/L and Q/R*). The first projection 'I' enters slightly more ventrally but originates laterally inside the LEa. Projection 'II' enters medially and slightly more dorsally. Both then construct a projection line ventrally to the ellipsoid body (*Figure 7K/L* 2 and Q/R; 'III'). Observing this pattern from a ventral view illustrates how these projection points lead to the actual ramifications inside the fan-shaped body (*Figure 7M–N*). Most importantly, the ramifications build a very thin and faint line in the non-pollen feeding outgroups (*Figure 7O/Q*), but a very prominent labelling of layer III of the fan-shaped body in *Heliconius* (*Figure 7P/R*). Whether these ER neurons solely branch in the fan-shaped body, as shown for GU neurons elsewhere (*Homberg et al., 2018*) or have additional side branches entering the ellipsoid body is not clear.

To carry visual information to the ellipsoid body, ER neurons arborize with neurons projecting from the AOTU in a conspicuous and small neuropil called the bulb (*Figure 7—figure supplement 2*). We wanted to examine whether divergence in ER neuron number and changes in fan-shaped body projections are concurrent with changes in bulb size, and potentially bulb microglomeruli number (*Figure 7—figure supplement 2*). We generated high-resolution images of the bulb to determine its size (*Figure 7—figure supplement 2C–F*), and 3D segmented seven microglomeruli per individual with which we generated an extrapolated approximation of total microglomeruli number by dividing bulb volume with average microglomerulus volume. This was necessary as most microglomeruli were not discernible from each other (*Figure 7—figure supplement 2G–H*). Importantly, we did not detect any significant species, nor sex differences in this dataset (*Figure 7—figure supplement 2I–K*). In addition, ER neuron number did not predict bulb size, or microglomeruli measures (*Supplementary file 1*).

In summary, we observed a conserved origin and projection pattern of GABA-ergic ER neurons in the ellipsoid body and a conserved size and microglomerular count in the bulb, but did detect a divergence in the number of ER neurons and corresponding size of their innervation in fan-shaped body layer III.

## Variation in Allatostatin A staining indicates species differences in NO and central body layers

Neuropeptides are often co-expressed with neurotransmitters to generate flexibility in neural circuits (*Kahsai and Winther, 2011*; *Nässel and Homberg, 2006*). Allatostatin A is a prominent example of these short peptides, where co-expression with serotonin, for example, might generate such flexibility in the response to sensory cues (*Vitzthum et al., 1996*). We labelled Allatostatin A expression with a Dip-allatostatin antibody, which revealed labelling of the neuron terminals near the synapses, as this is where the neuropeptides are transported to upon translation. Due to the species-dependent (*Pfeiffer and Homberg, 2014*) and state-dependent nature of neuropeptide expression and potential variation in expression levels (*Wegener and Chen, 2022*; *Schoofs et al., 2017*), we examined Allatostatin A expression in five different species (*H. melpomene*, *H. erato*, *Heliconius doris*, *Dryas iulia*, and *Dione juno*) and three to six individuals each (*Figure 8A*), with a consistent protocol throughout.

Across all species and individuals, we consistently identified an absence of staining in the protocerebral bridge and a very strong labelling in layers II and III of the fan-shaped body, with particularly high expression levels in layer II (*Figure 8B*). However, we observed considerable species-consistent variation in the remaining layers of the fan-shaped body, ellipsoid body and noduli (*Figure 8C*):

A. In the fan-shaped body we found broad and consistent expression in layers II and III across *Heliconius* and outgroups (*Figure 8D*). In *Heliconius doris,* Allatostatin A was also expressed in layer IV, with slight variation in terms of strength between the five individuals (compare a to b in *Figure 8D*).

B. In the ellipsoid body, we only detected strong expression in an area likely to be layers II and III in *Dione juno* and *Dryas iulia*, which are outgroup genera and do not feed on pollen, while we never saw strong expression in *Heliconius* (compare c to d in *Figure 8E*). In some cases, there was no expression at all (*Heliconius melpomene*), while in the case of *Heliconius erato* and *H. doris*, we only detected very weak and inconsistent expression (arrowheads in *Figure 8E*). Hence, these inter-specific differences reflect the divergence between pollen-feeding *Heliconius* and outgroups (*Figure 8C*).

C. Mirroring variation in general anatomy (*Figure 4C–D*), expression of Allatostatin A in the noduli varied across species. We only saw prominent expression in *Heliconius* (*Figure 8F–e*). Outgroup Heliconiini species always showed an absence of Allatostatin A expression (*Figure 8F–g*) or, in the case of *Dione juno,* faint expression in layer II of the noduli was observed in only two of five individuals. Again, any pronounced differences were linked to the split between pollen-feeding *Heliconius* and outgroups (*Figure 8C*).

We also have indications of frequent patterns of co-expression between neuropeptides and neurotransmitters (*Kahsai and Winther, 2011*; *Nässel and Homberg, 2006*), as Allatostatin A and Serotonin were expressed in fan-shaped body layers II and III (compare *Figure 8B* to *Figure 6F*), and dopamine in layer II (*Figure 6C*). The expression patterns we observed in Heliconiini are partially conserved when compared to *Drosophila*, and potentially with expression patterns in *Schistocerca* (*Kahsai and Winther, 2011*; *Vitzthum et al., 1996*). Moreover, *Kaiser et al., 2022* identified Allatostatin A expression in three fan-shaped and two ellipsoid body layers in the honey bee brain, which seems particularly similar to the expression patterns that we observed in *Dryas* and *Dione*. While they state that the ellipsoid body expression patterns seem to originate from populations of tangential neurons, the fan-shaped body expression pattern we observe in Heliconiini is very likely generated by columnar neurons (*Kaiser et al., 2022*). In our data, the columnar neurons were indeed labelled in some individuals, although it was not possible to connect labelled cell bodies to their resulting expression in the central body.

In sum, we identified pronounced variation in Allatostatin A expression, which contrasts with the conserved patterns observed for serotonergic and dopaminergic neurotransmitters. Interestingly, some of these differences were consistent between multiple *Heliconius* species and multiple outgroups, potentially reflecting differences that occurred alongside the origin of pollen feeding, derived spatial foraging behaviours, and mushroom body expansion.

## Discussion

*Heliconius* butterflies possess a unique dietary innovation, associated with derived foraging behaviours that are dependent on spatial memory and navigation. This evolutionary shift in learning and memory has previously been linked to a major expansion and specialisation of the mushroom body (*Farnworth et al., 2024a*; *Couto et al., 2023*; *Young et al., 2024*), but its impact on downstream pathways that must interact with mushroom body outputs to coordinate navigation and goal orientation behaviours has not been assessed previously. In this study, we identified that the volumetric expansion in the mushroom bodies in *Heliconius* butterflies did not occur in concert with volumetric changes in the central complex neuropils, AOTU, or POTU (*Figures 1 and 2*), which provide this function. We determined a largely conserved pattern of substructures (*Figures 4–6*), but did identify evidence of subtle, but highly specific patterns of divergence in the central complex of *Heliconius*, with an increased number of ER neurons, which innervate a specific layer of the fan-shaped body (*Figure 7*), and different expression patterns of the neuropeptide Allatostatin A, particularly in the noduli (*Figure 8*). Below, we discuss the potential functional and evolutionary implications of this overall conservation, and specific instances of divergence.

### Evolutionary implications of volumetric conservation

While the lack of volumetric change inside the central complex neuropils, as well as the AOTU and POTU, contrasts dramatically with mushroom body evolution in Heliconiini, it is more consistent with

the overall impression that the central complex has a largely conserved structure across all pyterygote insects (*Heinze, 2024*). Conservation of gross anatomy may indicate that it is not necessary to change the central complex, AOTU, and POTU substantially to modify cognitive processes in such a way to allow spatial foraging to occur. Indeed, volumetric conservation is important as it indicates probable conservation in the number of overall synapses, and hints at conserved terminal axonal and dendritic branches as well as glial processes, which are important domains of adaptive change in neural circuitry (*Roberts et al., 2022*; *Farnworth and Montgomery, 2024b*; *Konstantinides and Desplan, 2025*). However, what these synapses do and to which cell types they belong is not always reflected in gross anatomy, which means that these systems might indeed be meaningfully modified, just not in ways that are reflected by volumetric differences (*Farnworth and Montgomery, 2024b*; *Logan et al., 2018*; *Chittka and Niven, 2009*). In effect, any inferences of volumetric conservation may obscure many more fine-scale differences, such as some identified here (*Figures 7 and 8*), but potentially many more that have yet to be examined in an evolutionary framework (*Roberts et al., 2022*; *Farnworth and Montgomery, 2024b*; *Chittka and Niven, 2009*).

## Implications of central complex circuit logic on evolutionary constraints and adaptability

Evolutionary constraints, which are any aspect of a system's biology that limits the response to selection, are hypothesised to shape brain and neural system evolution by determining the landscape of adaptability in those systems (*Montgomery et al., 2016b*). While developmental constraints imply that conserved developmental programs reduce the available potential outcomes, functional constraints imply that in order to fulfil essential and ancestral functions, large parts of many neural circuitries must be conserved (*Montgomery et al., 2016b*). The considerable data available on the circuitry of the insect central complex offers possible explanations about how a dual pattern of conservation and divergence inside the central complex may come about.

The central complex is mostly made up of two dominant cell groups: columnar and tangential neurons. Each contains various subgroups with specific projection patterns, with other multicolumnar neuron types representing a relatively small proportion of cells (*Pfeiffer, 2023*; *Heinze, 2024*). Columnar neurons offer insights into developmental as well as functional constraints, while tangential neurons may be interpreted as this neural system's main axis of adaptability. Columnar neurons interconnect the different central complex neuropils and connect to the lateral complex (*Heinze and Homberg, 2008*; *Heinze et al., 2013*; *El Jundi et al., 2018*; *Hensgen et al., 2021*), where specific columnar types (PFL neurons) transfer information to descending pathways that relay navigational motor responses to the thoracic ganglia. Importantly, through their small projection fields, they innervate only small portions of the protocerebral bridge, fan-shaped body and ellipsoid body, and thus divide the central complex into vertical columns that, together with horizontal layers, create an orthogonal array of subdivisions (*Heinze and Homberg, 2008*; *Heinze et al., 2013*; *El Jundi et al., 2018*; *Hensgen et al., 2021*). This array is not only essential to central complex circuitry but also offers a very clear example of how structure is translated into function. The head direction circuit, formed by neurons branching in the protocerebral bridge and the ellipsoid body, illustrates this point particularly well. Here, many circuit elements are conserved even when structures vary markedly in shape, such as between toroid or bar-shaped ellipsoid bodies (*El Jundi et al., 2015*; *Stone et al., 2017*; *Hulse et al., 2021*; *Heinze and Homberg, 2007*; *Heinze and Homberg, 2008*; *El Jundi et al., 2018*; *Hensgen et al., 2021*; *Hulse and Jayaraman, 2020*; *Nguyen et al., 2021*; *Pisokas et al., 2020*), because each vertical column represents an equal portion of an animal's 360° space (*Seelig and Jayaraman, 2015*; *Hulse et al., 2021*; *Giraldo et al., 2018*). This universally required arrangement imparts cognitive demands on this circuit, thereby placing functional constraint on possible axes of variation, such as projection patterns, ramification size, and synaptic connectivity. This ultimately limits the variability of particular types of columnar neurons. Moreover, all columnar neurons are, per hemisphere, constructed by only four neural lineages, called DM1-4. These neural lineages seem to be largely conserved across insects, even in terms of absolute cell number (*Sayre et al., 2021*; *Farnworth et al., 2020*; *Walsh and Doe, 2017*; *Andrade et al., 2019*) which indicates considerable constraint in their developmental programs, and also implies that modifying the number of columnar neurons in a substantial way would neither be necessary nor useful to retain functionality. Instead, changes in the relative

distribution of specific neuron types that occupy each column may be a more functionally malleable axis of variation across species.

The adaptability of central complex function may instead be more likely to occur in tangential neurons (*Pfeiffer, 2023*; *Heinze, 2024*), such as the ER neurons that we characterised more closely. Tangential neurons connect other brain areas and other subcircuits to the central complex, thereby dividing the central complex into horizontal layers. These neurons may harbour more adaptability as they essentially 'plug in' information of context-dependent cues, such as different sensory modalities and internal states, from other brain regions, including the mushroom bodies, to the conserved array of columnar neurons (*Hulse et al., 2021*; *Kandimalla et al., 2023*; *von Hadeln et al., 2020*). Variability in these connections is also reflected in a broader developmental basis, where 18 different neural lineages are involved in the generation of tangential neurons (*Kandimalla et al., 2023*). As these neurons have a much more diverse developmental origin, location in the brain and independent function (*Kandimalla et al., 2023*; *von Hadeln et al., 2020*), we speculate that input strength, ramification size and synaptic changes can occur partially independently of the columnar neurons.

Layers that are formed through these tangential neurons, particularly in the fan-shaped body, are thought to be one of the most variable domains in central complex circuitry (*Heinze, 2024*), for which we here offer a rare comparative example by identifying an increased innervation of fan-shaped body layer III in *Heliconius* butterflies, which rely on spatial memory to direct foraging behaviour (*Figure 7*). Indeed, the fan-shaped body at large is involved in selecting and generating goal directions, and a subsequent comparison to signals encoded by the head direction circuit will then result in a specific navigational response (*Mussells Pires et al., 2024*; *Westeinde et al., 2024*; *Hulse et al., 2021*). To allow navigation to a specific goal, the fan-shaped body receives information related to translational movements via the noduli (*Stone et al., 2017*; *Lyu et al., 2022*; *Lu et al., 2022*), as well as a multitude of other direct and indirect inputs from mushroom body output neurons for example, which encode memory-specific signals and other internal state information (*Matheson et al., 2022*; *Kandimalla et al., 2023*; *Donlea et al., 2018*). As such, it may vary more across species, reflecting the specific behavioural repertoires of a lineage, to provide integration that often depends on shifts in sensory cues and increased reliance on internal motivation *Heinze, 2024*, which may be closely linked to ecological need. Indeed, a columnar architecture is often less obvious in the fan-shaped body than in the ellipsoid body and protocerebral bridge (*El Jundi et al., 2018*; *Hensgen et al., 2021*), potentially reflecting its necessary functional variability (*Hulse et al., 2021*; *Sayre et al., 2021*; *Jahn et al., 2023*; *Heinze, 2024*).

## Implications of fine differences in a conserved central complex circuit

Our neuromodulator labelling permitted us to take a closer look at potential divergent traits, and interestingly these match to the areas of the central complex where adaptive variation could be expected to occur, that is the fan-shaped body and noduli (*Hulse et al., 2021*; *Heinze, 2024*). While significant parts of the neurotransmitter system in Heliconiini seem conserved (*Figure 6*), matching previous observations (*Farnworth et al., 2024a*), we identified quantitative differences in GABAergic ER neurons, as these occurred in higher number in *Heliconius* and produced an increased innervation of fan-shaped body layer III (*Figure 7*). These results shine light on the much less examined portion of ER neurons that innervates the fan-shaped body but in a direct comparative setting. *Drosophila* connectome data (*Hulse et al., 2021*; *Kandimalla et al., 2023*), as well as immunohistochemical data across insects (*Homberg et al., 2018*), indicate that the DALv2 lineage and the resulting ER neuron population always have a conserved portion of cells that connect to the fan-shaped body, but potentially varying amounts relative to the whole ER neuron population across insects, even if this is only 5% in the case of *Drosophila* (Figure 1 in *Kandimalla et al., 2023*). Indeed, the projection pattern of the fan-shaped body projecting neurons in *Heliconius melpomene* closely resembles their homologous counterparts in *Drosophila*, despite their distant phylogenetic relationship (Figure S4 in *Kandimalla et al., 2023*). Our data also indicate that this portion of cells could be subject to a significant amount of evolutionary adaptability particularly in terms of cell numbers. This is corroborated by substantial differences in fan-shaped body ramification patterns across insects (*Homberg et al., 2018*).

In addition to the ER population and its arborisations in the fan-shaped body, we also closely examined the bulb (*Figure 7—figure supplement 2*). The bulb is made up of large synaptic complexes, or microglomeruli, constructed primarily by TuBu neurons that enter the bulb from the AOTU and

ER neurons that connect to these TuBu neurons and project any modified signals into the ellipsoid body (*El Jundi et al., 2018*; *Mota et al., 2016*; *Held et al., 2016*; *Schmitt et al., 2016*; *Träger et al., 2008*). We did not identify any species differences in several variables of bulb anatomy (*Figure 7—figure supplement 2*), despite an increased number of ER neurons. This supports our hypothesis that the portion of ER neurons that is increased in *Heliconius* might partly be those which project to the fan-shaped body, without arborising inside the bulb at all. Alternatively, these ER neurons would still arborise in the bulb, but to the same number of TuBu neurons, thus not requiring increased number of synapses which might be reflected in the variables that we examined, but then branch into the fan-shaped body. Either way, our finding of conserved bulb architecture narrows down the area of change considerably.

This GABA-ergic ER neuron population and related projection differences, despite their relatively subtle appearance, may have functional and behavioural relevance. In a recent model (*Le Moël et al., 2019*), 'vector memory neurons' synapse onto and inhibit valence signals in fan-shaped body columnar neurons. Through this connection, a current heading direction could be inverted, inducing the return of a path-integrating animal to its original position (*Le Moël et al., 2019*). This could be an important element in the context of navigational demands placed on *Heliconius* butterflies, who need to reliably memorise and navigate to and from a nocturnal roost site during their spatial foraging navigation (*Young and Montgomery, 2020*; *Finkbeiner, 2014*). An increased portion of inhibitory vector memory cells, in this case ER neurons projecting onto 'integrator neurons' in fan-shaped body layer III, could fulfil these demands in a more precise, more parcellated manner.

In addition, we also detected substantial, but species-specific, variation in Allatostatin A labelling in the central complex. Across the insect brain, Allatostatin A has broad roles in the regulation of feeding, development, alternation of activity and sleep as well as learning (*Wegener and Chen, 2022*; *Kreissl et al., 2010*; *Urlacher et al., 2016*; *Yamagata et al., 2016*; *Suárez-Grimalt and Raccuglia, 2021*). We described four patterns of variation (*Figure 8C*), two of which are consistent with a divide between the pollen feeding *Heliconius* and outgroup genera. While some of these differences are difficult to interpret, the expression of Allatostatin A inside large parts of the noduli in *Heliconius* is intriguing. The noduli are the target of several tangential neurons, termed TN1 and TN2, carrying information about self-motion cues such as translational optic flow to estimate distance (*Stone et al., 2017*; *Lyu et al., 2022*; *Lu et al., 2022*). It is possible that Allatostatin A is expressed alongside neurotransmitters to co-regulate the activity of specific populations of such tangential neurons or, indeed PFN neurons that receive information by these neuron populations and then project these to the fan-shaped body. Interestingly, the central complex of bumblebees, like *Heliconius*, an allocentric spatial forager, houses twice the amount of PFN neurons than the central complex of fruit flies (*Sayre et al., 2021*). This could improve spatial foraging by integrating motion and locomotory information with compass information to induce novel goal directions (*Stone et al., 2017*). Moreover, spatially foraging Hymenoptera harbour modifications of their noduli, a 'nodulus cap', likely being a unique structural adaptation to specific computational needs surrounding navigation (*Sayre et al., 2021*; *Heinze, 2024*). Hence, while there is currently no direct evidence indicating how the behavioural and cognitive challenges associated with long-term spatial memory in *Heliconius* are functionally connected to the changes in Allatostatin A expression in the central complex (*Figure 8*), there is comparative evidence in other foragers suggesting this site may be functionally important for spatial behaviours.

## Information pathways for spatial foragers into the central complex

Like in other insects (*Homberg et al., 2018*), GABAergic ER neurons in Heliconiini butterflies mostly have synaptic input in the bulb and transfer information into the central complex. ER neurons projecting to the ellipsoid body are part of a visual compass pathway and encode external orientation cues, such as the sun azimuth, polarised skylight information and panoramic scenes (*Heinze and Reppert, 2011*; *Nguyen et al., 2022*; *Nguyen et al., 2021*), information that would be especially relevant during

spatial foraging of *Heliconius* butterflies. ER neurons carry visual information to the ellipsoid body, where they inhibit the head direction network (*Fisher et al., 2019*). Through Fas II staining (*Figure 6*), we were able to identify neurons that interconnect the ellipsoid body with the gall and the protocerebral bridge, and that are likely the homologues of EPG neurons in the Heliconiini central complex. Thus, like in other insects, the Heliconiini central complex is equipped with a head direction network that encodes a current head direction of the butterflies. Compass information of these EPG neurons is likely transmitted in the protocerebral bridge via Δ7 neurons to PFN neurons, a type of neuron that has also been reported in several insects, including butterflies (*Heinze and Homberg, 2008*; *Heinze et al., 2013*; *El Jundi et al., 2018*). Fruit fly PFN neurons are involved in transforming the heading signal into a body-invariant travelling direction in the fan-shaped body (*Lyu et al., 2022*; *Lu et al., 2022*). Lepidopteran PFN neurons likely have the identical function.

Importantly, we also found that some ER neurons bypass the ellipsoid body and give rise to dense branches within distinct layers in the fan-shaped body (ER-FB) and an increased innervation in spatially foraging *Heliconius*. In addition, our GAD staining also revealed GABA-immunoreactivity in arborisations in the SMP, a major target region of mushroom body output neurons (MBONs) carrying modified synaptic input from Kenyon cells to other areas of the brain (*Li et al., 2020*). Although we cannot undoubtedly assign these GABA-ergic arborisations to the ER-FB neurons, additional branches of homologous cells have been previously described for *Drosophila* ER-FB neurons (*Kandimalla et al., 2023*). It is therefore plausible that specific mushroom body output signals of the massively enlarged mushroom bodies of *Heliconius* species connect to an increased portion of ER-FB neurons, subsequently projecting into the fan-shaped body. Indeed, through dye injections, we also found that neurons of the SMP/SLP project to the fan shaped body (*Figure 5*), similar to what has been described in the Monarch central complex (*Heinze et al., 2013*).

In addition to its classic attributes of associative learning and memory, mushroom bodies produce and encode familiarity of visual scenes, which is an essential information if visual scenes and olfactory signals are used to associate specific environments with a 'homing vector' during trap-lining or other forms of spatial foraging. How familiarity is integrated into the navigational circuit of the central complex is largely unknown (*Collett et al., 2025*) but our results support recent models suggesting that familiarity cues from the SMP/SLP could provide valence information based on memorised visual scenes and olfactory cues that gate goal direction navigation in the fan-shaped body, a mechanism proposed to exist in *Drosophila* (*Matheson et al., 2022*). In this context, ER-FB neurons could play a major role in modulating the signals of MBONs in the SMP/SLP and/or in the fan-shaped body, before they are used to produce steering signals in the fan-shaped body. Such a mechanism would allow *Heliconius* butterflies to store and recall a number of vectors in the fan shaped body and, thus, would establish the neural correlate of a navigation strategy akin to trap-lining (*Young and Montgomery, 2020*).

## Conclusion

By leveraging the unique biology of *Heliconius* butterflies and their close phylogenetic relationships with outgroups that lack their behavioural specialisations, we provide a rare example of evolutionary divergence of the central complex in fine-anatomical detail among closely related species. Our study has identified areas of the central complex circuitry where changes may occur to modify circuitry in such a way to facilitate the evolution of spatial foraging. The central complex is an intriguing system to study circuit divergence, exactly because it is generally so highly conserved, and because its structure is directly translated into its function (*Heinze, 2024*). In light of this general conservation, any fine differences that are identified are potentially easier to connect to specific behavioural traits compared to more divergent circuits. The model system of spatial foraging adaptation in Heliconiini butterflies highlights very different ways to modify circuitries in the two integration and navigation centres, the mushroom body and central complex (*Farnworth et al., 2024a*; *Collett et al., 2025*). It is therefore uniquely suited to be examined further to investigate the precise circuitry underpinnings of constraints and adaptability (*Farnworth and Montgomery, 2024b*).

## Materials and methods

**Key resources table**

| Reagent type (species) or resource | Designation | Source or reference | Identifiers | Additional information |
|---|---|---|---|---|
| Biological sample (Heliconiini butterflies) | Binomial, standard | The Entomologist; Costa Rica Entomological Supply | N/A | See Materials and methods for details |
| Antibody | Anti-HRP (rabbit; polyclonal) | Merck Sigma Aldrich | Cat# P7899, RRID:AB_261181 | IF (1:5000); *Supplementary file 2* |
| Antibody | Anti-acTUB (mouse; monoclonal) | Merck Sigma Aldrich | Cat# T7451, RRID:AB_609894 | IF (1:100); *Supplementary file 2* |
| Antibody | Anti-SYNORF1 (mouse; monoclonal) | DSHB | Cat# 3C11, RRID:AB_528479 | IF (1:20; Batch 68 ug/ml); *Supplementary file 2* |
| Antibody | Anti-FASII (mouse; monoclonal) | DSHB | Cat# fasciclin II 2F5, RRID:AB_10805878 | IF (1:20; Batch 30 µg/ml); *Supplementary file 2* |
| Antibody | Anti-GAD (rabbit; polyclonal) | Merck Sigma Aldrich | Cat# G5163, RRID:AB_477019 | IF (1:500); *Supplementary file 2* |
| Antibody | Anti-TH (rabbit; polyclonal) | Millipore | Cat# AB152, RRID:AB_390204 | IF (1:200); *Supplementary file 2* |
| Antibody | Anti-5-HT (rabbit; polyclonal) | Immunostar | Cat# 20080, RRID:AB_572263 | IF (1:400); *Supplementary file 2* |
| Antibody | Anti-Allatostatin-A (rabbit; polyclonal) | Jena Bioscience | Cat# ABD-062, RRID:NA | IF (1:500); *Supplementary file 2* |
| Antibody | Anti-mouse-Cyanine-2 (goat; polyclonal) | Jackson Immunoresearch | Cat# 115-225-146, RRID:AB_2307343 | IF (1:100); *Supplementary file 2* |
| Antibody | Anti-mouse-Cyanine-3 (goat; polyclonal) | Jackson Immunoresearch | Cat# 115-165-003, RRID:AB_2338680 | IF (1:100); *Supplementary file 2* |
| Antibody | Anti-rabbit-Cyanine-2 (goat; polyclonal) | Jackson Immunoresearch | Cat# 111-225-144, RRID:AB_2338021 | IF (1:100); *Supplementary file 2* |
| Antibody | Anti-rabbit-Cyanine-3 (goat; polyclonal) | Jackson Immunoresearch | Cat# 111-165-144, RRID:AB_2338006 | IF (1:100); *Supplementary file 2* |
| Chemical compound, drug | Low melting-point Agarose | Thermo Fisher Scientific, MA, USA | Cat. #: 16520–050 | See Materials and methods for details |
| Chemical compound, drug | PBS | ThermoFisher Scientific, MA, USA | Cat. #: BR0014G | See Materials and methods for details |
| Chemical compound, drug | NGS | MERCK, Germany | Cat. #: G9023 | See Materials and methods for details |
| Chemical compound, drug | Fluoro-ruby | Thermo Fisher Scientific, MA, USA | Cat. #: D1817 | See Materials and methods for details |
| Chemical compound, drug | Methyl salicylate | MERCK Sigma Aldrich, MA, USA | Cat. #: M6752 | See Materials and methods for details |
| Chemical compound, drug | DAPI | Merck Sigma Aldrich | Cat. #: D9542 | *Supplementary files 1 and 2* :1000 |
| Software, algorithm | Amira 2020.2/5.4.3/2021.1 | Thermo Fisher Scientific *Stalling et al., 2005* | RRID:SCR_007353 | Effects of version and experimenter were ruled out statistically |
| Software, algorithm | R 4.3.1 | R Core Team | RRID:SCR_001905 | See *Supplementary file 1* for packages; see data repository for script |
| Software, algorithm | R Studio 2023.06.0 | Posit Team (*Posit team, 2023*). | RRID:SCR_000432 | See Materials and methods for details |
| Software, algorithm | Fiji 1.54 | Fiji Team; *Schindelin et al., 2012* | RRID:SCR_003070 | See Materials and methods for details |
| Other | Critical care formula | VETARK, Winchester, UK | N/A | See Materials and methods for details |

| Reagent type (species) or resource | Designation | Source or reference | Identifiers | Additional information |
|---|---|---|---|---|
| Other | Frosted object slides | Thermo Fisher Scientific, MA, USA | J1800AMNZ | See Materials and methods for details |

We have derived our results from two datasets with different origins, which we describe below for each. Data which is relevant to figures and statistics is deposited at https://doi.org/10.5281/zenodo.15304965.

### Dataset for large-scale volumetric analyses

All the 307 individual Heliconiini butterflies used in the large-scale statistical analysis (*Figures 2 and 3*, *Figure 2—figure supplements 1–3*) were wild-caught and are part of the same dataset used in a previous study (*Couto et al., 2023*). Here, we segmented the neuropils of the central complex (CX) as well as the anterior (AOTU) and posterior optic tubercle (POTU) and used the openly available data for the mushroom body and other measures to generate a collective volumetric dataset for 307 individuals of 41 species of Heliconiini for the CX, AOTU, and POTU as well as the mushroom body (MB) and central brain volume (CBR), the basis for the allometric control variable. Note that we did not have noduli (NO) sizes for all individuals due to some inconsistent image quality, which is why we excluded NO size in the within neuropil analysis as predictor of all others and only used it as dependent variable (missing values in *Figure 3*).

### Dataset for small-scale fine-anatomical analyses

Individuals used for finer scale data analysis (*Figures 4–6 and 8*, *Figure 7—figure supplements 1 and 2*) overlapped partially with butterflies used previously (*Farnworth et al., 2024a*), and as a result, rearing procedures, fixations, and stainings were highly similar or identical. In short, pupae of Heliconiini butterflies were ordered from commercial suppliers (The Entomologist; https://butterfly-pupae.com/ or Costa Rica Entomological Supply; www.butterflyfarm.co.cr). They were then attached posteriorly to a microfibre cloth in a pop-up cage. After eclosion, they were given IDs, marked on their forewings, to later identify their age when sampling. They were reared in 2x2 x 2 m mesh cages at 26 °C, 80% humidity and a 16 hr/8 hr light/dark cycle, with ad libitum feeding consisting of flowering *Lantana* plants as pollen and nectar supply and artificial feed containing 20% sugar and 5% critical care formula (VETARK, Winchester, UK) in water. All images displayed in figures are supplied at https://doi.org/10.5281/zenodo.15304965, with a list of which figure refers to which file in *Supplementary file 2*.

### 3D segmentations of volumetric dataset

3D segmentations of the CX, its neuropils as well as the AOTU and POTU were performed using Amira Version 2020.2 and Version 5.4.3 (effects of version and experimenter were ruled out statistically) (*Stalling et al., 2005*). For the segmentations, we used the identical procedure as previously described (*Couto et al., 2023*). First, to correct for axial aberrations due to differences in the refractive indices of air and the mounting medium of the samples of the volumetric dataset, we used a corrective scale factor of 1.52 on the z-dimension as determined previously (*Montgomery et al., 2016a*). We then used the segmentation editor on a grey image stack of labelled brain to mark each neuropil through the brush or wand tool in approximately every 10 slices for larger neuropils and 3–5 slices for smaller ones, using interpolation subsequently. Manual segmentation and interpolation was verified using all views (xy, yz, xz) of the segmentation editor. The selection was smoothed in each view for most neuropils, except for the small neuropils, that is the POTU and NO. Smoothing of the 3D surface of reconstructed neuropils was only performed for visual representations such as in *Figure 1*. Here, other neuropils, such as the mushroom body, antennal lobe, and visual neuropils were segmented identically as described in *Montgomery et al., 2016a*.

### Statistical tests of volumetric dataset

All tests were performed with R 4.3.1 in R Studio 2023.06.0 (*Posit team, 2023*; *R Development Core Team, 2023*). Packages used and full results are found in *Supplementary file 1*, while the script is found at https://doi.org/10.5281/zenodo.15304965. To assess patterns of variation in this species-rich dataset we used phylogenetically corrected GLMMs (general linear mixed models) based on Bayesian

statistics using *mcmcglmm* (*Hadfield, 2010*). We always ran every model twice and compared effects by using the Gelman Diagnostics criterion. We checked for their common convergence through analytical as well as visual means and determined covariance values below 1.1 as acceptable. Priors were set for residuals and random effects with a univariate inverse-Wishart distribution (V=1) and a degree of belief parameter (nu) of 0. We sampled 500,000 iterations with a burnin of 10,000 and sampling each 500 iterations. We used the phylogenetic tree published with *Couto et al., 2023*. We compared to null models when testing the effects of interactions of pollen feeding/clade membership to volumes using the difference and extents of DIC. In all cases, if the DIC of the test model was larger than null, the differences were miniscule.

To assess differences in evolutionary rate in CX, AOTU, and POTU compared to what has been determined in the MB by *Couto et al., 2023*, we used an identical approach. We calculated species averages and used a multirate Brownian model assumption to calculate rates. CX rates were regressed to rCBR rates. Residuals were then plotted with colour coding according to the original MB plot to assess MB and CX differences comparatively.

To assess differences within tested structures (CX neuropils, AOTU, POTU), we performed *mcmcglmm* models with one structure as dependent variables and all others as independent variables. In case of the NO, we only used data points where NO data was available, thus reducing the amount of data points tested. To account for multiple testing effects in these within-effects tests, we multiplied the p value by four for the main within tests as each structure was included four times in other tests. For the *posthoc* effects tests of the significant results, we multiplied p values by the number of significant relationships, that is 10.

## Antibody use and characterisation for fine-scale analysis

All antibodies used in this study are described with information about origin and usage in *Supplementary file 2* and the Key Resources Table. All antibodies used are commercially available, their specificity carefully assessed and widely established and have revealed conserved expression patterns previously (*Timm et al., 2021*; *Homberg et al., 2023b*; *Homberg et al., 2018*; *Vitzthum et al., 1996*; *Kreissl et al., 2010*). We used synapsin, acetylated tubulin, and HRP as structural markers. The combination of tubulin and HRP reveals the whole neuronal structure, from cell soma to synaptic endings (*Farnworth et al., 2024a*). We selected a series of markers for neuromodulators that are typical labels of fine structures in the CX *Pfeiffer and Homberg, 2014* to reveal series of specifically labelled neurons. These were antibodies against Glutamate decarboxylase (GAD) to label the neurotransmitter γ-aminobutyric acid (GABA), Tyrosine hydroxylase (TH) to label the neurotransmitter dopamine, Serotonin or 5-hydroxytryptamine (5-HT) and the neuropeptide Allatostatin-A. We also used a Fasciclin-II antibody as selective label only, as it previously revealed specific populations of neurons relevant to the CX (*Farnworth et al., 2024a*).

## Dissection, fixation, immunostaining, and bulk injections for fine-scale analysis

These procedures were close to identical to our previous study (*Farnworth et al., 2024a*) and are highly similar to published protocols (*Couto et al., 2023*; *Ott, 2008*). Butterflies with ages ranging from 3 to 5 days were decapitated following a short cold-anaesthesia. Their antennae, proboscis and palps were removed and the head was pinned with anterior to the top. First, the head was opened by removing the interocular cuticle up to and including the antennal roots. We then exposed the head and brain to the fixative in situ, fixed the brains for 22 hr at room temperature on an orbital shaker, and only subsequently, removed the brain from the head capsule. We used HBS (HEPES-buffered saline; 150 mM NaCl; 5 mM KCl; 5 mM CaCl$_2$; 25 mM sucrose; 10 mM HEPES) as dissection medium. We used 1% Zinc-Formaldehyde (ZnFA; 18.4 mM ZnCl$_2$, 135 mM NaCl, 35 mM sucrose, 1% formalin) as fixative, as described previously (*Ott, 2008*), having a less stark effect of introducing aberrations in the tissue. After fixation and dissection, all brains were subjected to a 2 hr incubation in Dent's solution (80% Methanol, 20% DMSO) with a subsequent rinse and storage in methanol at –20 °C. We rehydrated brains in a descending methanol series (90%, 70%, 50%, 30%) diluted in 0.1 M Tris-HCl (pH = 7.4), by washing them in each dilution for 10 min, and a final wash in 100% 0.1 M Tris-HCl.

We sectioned brains at 80 µm using a Leica Vibratome VT1000-S (Leica Biosystems, IL, USA) with speed and frequency set to 5. The brains were prepared for sectioning by embedding the rehydrated

brains in 5% low melting-point Agarose (#16520–050, Thermo Fisher Scientific, MA, USA) and fixing them on a magnetic disk holder using cyanoacrylate glue, maintaining a non-tilted anterior-posterior vertical axis as much as possible.

Immunostaining was performed as reported previously. We used PBS-d (0.1 M PBS [BR0014G, Thermo Fisher Scientific, MA, USA] with 1% DMSO) as wash buffer, and started with a rinse subsequent to sectioning. We then performed a 30-min permeabilisation wash with permeabilisation buffer (PBS-d and 2% Triton-X-100), rinsed once again with PBS-d and then started blocking using 5% of NGS (normal goat serum, G9023, MERCK, Germany) in PBS-d for 2–4 hr. We then applied the first antibodies with the appropriate dilution (see *Supplementary file 2*) in blocking buffer and incubated them for 3 days at 4 °C on an orbital shaker. We then rinsed the brain sections once and washed them three times, for 30 min each, with PBS-d. Secondary antibodies were diluted in blocking buffer (see *Supplementary file 2*) and incubated for 3 days at 4 °C on an orbital shaker. This was followed by a rinse in PBS-d and a 45-min incubation of DAPI (1:1000) in water with 0.2% Triton-X-100. This was followed again by a PBS-d rinse and four 10-min washes, and a final wash in PBS for 1 hr. We then transferred brains to 60% glycerol in PBS overnight for approximately 12–16 hr at 4 °C or 2–4 hr at room temperature. Brains were then transferred to 80% glycerol for an hour and mounted on frosted object slides (J1800AMNZ, Thermo Fisher Scientific, MA, USA) covered with #1.5 size coverslips and sealed with nail polish.

In addition to immunostainings, we also made use of a dextran injection into the superior medial and lateral protocerebrum (*Figure 5*). This injection followed procedures described elsewhere (*Couto et al., 2023*) and were exactly performed as in *Farnworth et al., 2024a*. Butterflies were secured in custom-made holders having a plastic collar around the cervix, with a waterproof barrier created by using low melting point wax around the head to prevent leakage of ringer solution (composition: 150 mM NaCl, 3 mM CaCl$_2$, 3 mM KCl, 2 mM MgCl$_2$, 10 mM HEPES, 5 mM Glucose, 20 mM Sucrose). We then exposed the dorsal region of the brain under the ringer solution and under filtered illumination, inserted the tip of a pulled glass capillary (G100-4, Warner Instruments, CT, USA), loaded with crystals of fluoro-ruby (dextran-tetramethylrhodamine: 10,000 MW, D1817, Thermo Fisher Scientific, MA, USA) mixed in 2% bovine serum albumin (BSA), into the cortical region of the brain. The capillary tip was held inserted for a few seconds. Subsequently, the head was covered with fresh ringer solution and kept overnight in a dark and humid chamber to facilitate dye diffusion. After incubation, the brain was dissected out, fixed in ZnFA, and immunostained with anti-SYNORF1 (3C11, Developmental Studies Hybridoma Bank, University of Iowa, IA, USA) following the same standard procedures described above and elsewhere (*Couto et al., 2023*; *Ott, 2008*). After immunostaining, the brain was exposed to a series of increasing glycerol concentrations (1%, 2%, 4% for 2 hr each, and 8%, 15%, 30%, 50%, 60%, 70%, 80% for 1 hr each) in 0.1 M Tris buffer with 1% DMSO, before complete dehydration with three consecutive washes in 100% ethanol (for 30 min each) and embedding in methyl salicylate (M6752, MERCK Sigma Aldrich, MA, USA).

## Imaging and image analysis for fine-scale analysis

We used a Leica SP8 (Leica Microsystems) and a 20 X air objective (20 X HC PL APO CS2, NA = 0.75) and for high-detail imaging a 40 X oil objective (40 x HC PL APO CS2, NA = 1.3) for the GABA-ergic labelling of the CX and a 60 X oil objective (63 x HC PL APO CS2, NA = 1.4) for bulb imaging. We applied a 65 mW Ar laser, a 20 mW DPSS yellow laser, and a 50 mW 405 nm diode laser to excite Cyanine-2 linked signal, Cyanine-3 linked signal, and DAPI, respectively. Fluorescence signal was scanned using Hybrid detectors for Cyanine-2/3 and a PMT detector for DAPI. We used line averaging of 2–3, while using bidirectional scanning at 600 Hz with a resolution of at least 1024 x 1024. To calculate pinhole sizes, we used either the average of all emission maxima in use or the smallest emission maximum and set AU = 1. We then applied the system optimised z-slice size and linear Z compensation where necessary. Using the edge we cut into the agarose gel before slicing, we identified the same hemisphere consistently.

## Image processing, 3D segmentation, and annotation for fine-scale analysis

To generate accurate 3D segmentations in our fine-scale anatomical analysis, we corrected for axial aberrations due to refraction index differences between air and the mounting medium of 80% glycerol

(*Bucher et al., 2000*) using a correction value of 1.22 determined previously when using the 20 X air objective (*Farnworth et al., 2024a*).

We used Fiji 1.54 (*Schindelin et al., 2012*) and the included standard tools to modify brightness/contrast and orientation in all axes. We used an identical procedure described in *Farnworth et al., 2024a* to generate a merged whole brain picture from brain slices, which in essence was based on creating concatenated stacks in Fiji with identical orientations, and then compensating for changes in both X and Y, using Amira 3D 2021.1 (Thermo Fisher Scientific, MA, USA) and its module *AlignSlices* in the first channel. Using *AlignSlices* on the second channel, we then used the first aligned channel as reference, enforcing identical shifts in X-Y in both channels.

The CX and associated parts in *Figures 4 and 7* were segmented using the *labelfield* module and manual segmentation in approximately every 5–10 slices (2.5–5.5 µm) combined with interpolation. Interpolated segmentations were corrected in all views available. The selection as well as the segmentation was smoothed (value of 4 in all axes). To generate a surface view, we used constrained smoothing at a level of 3.

To segment the bulb, we created high-resolution images and were particularly careful to only segment the area of the bulb that comprised large synapses/glomeruli, excluding parts of the LEa/IT projection. This was essential, because we relied on extrapolating the total number of microglomeruli from a subset of segmented microglomeruli and the total volume that contained microglomeruli, which means any section containing tracts and not glomerular structures would skew the estimated total number of microglomeruli. Extrapolation was necessary, as not all microglomeruli were visually discernible. We achieved an unskewed bulb volume by leaving out dense pieces of tubulin-positive tract material. We segmented seven microglomeruli per individual from the posterior section of the bulb, where they were most clearly visible, to get the most comparable impression across individuals and species. We then calculated average microglomerulus size and divided this by bulb volume to determine an approximation of microglomeruli number. To determine quantified patterns with these bulb metrics, we performed standard linear regressions in R. To determine ER neuron numbers in *Figure 7*, we counted cell bodies of one hemisphere manually, and performed standard linear regressions in R. We used Inkscape (https://inkscape.org) to generate all figures.

## Acknowledgements

We thank Antoine Couto for providing the mass injection data in *Figure 5*, and Francesco Cicconardi for providing image material of the butterfly icons used throughout. We further thank Barbara Webb and Stanley Heinze for pointing us to useful computational models in relation to the role of inhibitory ER neurons. We thank Christian Wegener for his insights into the Allatostatin A stainings. This project began under a NERC Independent Research Fellowship NE/N014936/1 and ERC Starter Grant 758508 and was completed under HFSP Project Grant RGP0029/2022 to SHM and BEJ. MSF was supported by a Walter-Benjamin Fellowship from the Deutsche Forschungsgemeinschaft (FA 1818/1-1). YPT is supported by an ERC grant (851040) to Dr Richard M Merrill (LMU), TL is supported by a University of Bristol Scholarship, and EH is supported by an MRC DTP studentship (MR/W006308/1).

## Additional information

### Funding

| Funder | Grant reference number | Author |
| --- | --- | --- |
| Natural Environment Research Council | Independent Research Fellowship NE/N014936/1 | Stephen H Montgomery |
| European Research Council | Starter Grant 758508 | Stephen H Montgomery |
| Human Frontier Science Program | Project Grant RGP0029/2022 | Basil el Jundi Stephen H Montgomery |
| Deutsche Forschungsgemeinschaft | Walter-Benjamin Fellowship FA 1818/1-1 | Max S Farnworth |

| Funder | Grant reference number | Author |
|---|---|---|
| University of Bristol | Student Scholarship | Theodora Loupasaki |
| Medical Research Council | DTP studentship MR/W006308/1 | Elizabeth A Hodge |
| European Research Council | Starter Grant 851040; to Richard M Merrill | Yi Peng Toh |

The funders had no role in study design, data collection and interpretation, or the decision to submit the work for publication.

### Author contributions

Max S Farnworth, Conceptualization, Resources, Data curation, Formal analysis, Funding acquisition, Validation, Investigation, Visualization, Methodology, Writing – original draft, Writing – review and editing; Yi Peng Toh, Theodora Loupasaki, Elizabeth A Hodge, Investigation, Writing – review and editing; Basil el Jundi, Conceptualization, Validation, Writing – original draft, Writing – review and editing; Stephen H Montgomery, Conceptualization, Formal analysis, Supervision, Funding acquisition, Validation, Writing – original draft, Project administration, Writing – review and editing

### Author ORCIDs

Max S Farnworth ⓘ https://orcid.org/0000-0003-2418-3203
Yi Peng Toh ⓘ https://orcid.org/0000-0001-7110-5270
Theodora Loupasaki ⓘ https://orcid.org/0009-0004-9066-224X
Elizabeth A Hodge ⓘ https://orcid.org/0009-0001-1837-4719
Basil el Jundi ⓘ https://orcid.org/0000-0002-4539-6681
Stephen H Montgomery ⓘ https://orcid.org/0000-0002-5474-5695

Reviewer #1 (Public review): https://doi.org/10.7554/eLife.107589.3.sa1
Reviewer #2 (Public review): https://doi.org/10.7554/eLife.107589.3.sa2
Author response https://doi.org/10.7554/eLife.107589.3.sa3

# Additional files

### Supplementary files

Supplementary file 1. Reporting of *R* packages, datasets, and results.

Supplementary file 2. Reporting of antibodies with additional detail and data sources for figures.

MDAR checklist

### Data availability

Data, R script and readme files associated with this publication is publicly available at https://doi.org/10.5281/zenodo.15304965.

The following dataset was generated:

| Author(s) | Year | Dataset title | Dataset URL | Database and Identifier |
|---|---|---|---|---|
| Farnworth MS | 2025 | Dataset associated with publication of Farnworth et al "Distinct evolutionary trajectories of two integration centres, the central complex and mushroom bodies, across Heliconiini butterflies" | https://doi.org/10.5281/zenodo.15304965 | Zenodo, 10.5281/zenodo.15304965 |

The following previously published dataset was used:

| Author(s) | Year | Dataset title | Dataset URL | Database and Identifier |
|---|---|---|---|---|
| Couto A, Young F, Atzeni D, Marty S, Melo-Flórez L, Hebberecht L, Monllor M, Neal C, Cicconardi F, McMillan W, Montgomery SH | 2023 | Data From: Rapid expansion and visual specialisation of learning and memory centers in the brains of Heliconiini butterflies | https://doi.org/10.5061/dryad.f1vhhmh28 | Dryad Digital Repository, 10.5061/dryad.f1vhhmh28 |

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
