## [Editor Report · eLife Assessment]

The analysis of neural morphology across Heliconiini butterfly species revealed brain area-specific changes associated with new foraging behaviours. While the volume of the centre for learning and memory, the mushroom bodies, was known to vary widely across species, these new, **valuable** results show conservation of the volume of a center for navigation, the central complex, but with specific changes in neuropeptide expression in the noduli and in the numbers of ellipsoid body ring neurons. The presented evidence is **convincing** for both volumetric conservation in the central complex and fine neuroanatomical differences associated with pollen feeding, delivered by experimental approaches that are applicable to other insect species. This work will be of interest to evolutionary biologists, entomologists, and neuroscientists.

---

## [Referee Report · Reviewer #1 (Public review)]

The authors previously reported that Heliconius, one genus of the Heliconiini butterflies, evolved to be efficient foragers to feed pollen of specific plants and have massively expanded mushroom bodies. Using the same image dataset, the authors segmented the central complex and associated brain regions and found that the volume of the central complex relative to the rest of brain are largely conserved across the Heliconiini butterflies. By performing immunostaining to label specific subset of neurons, the authors found several potential sites of evolutional divergence in the central complex neural circuits, including the numbers of GABAergic ellipsoid body ring neurons and the innervation patterns of Allatostatin A expressing neurons in the noduli. These neuroanatomical data will be helpful to guide the future studies to understand the evolution of the neural circuits for vector-based navigations.

Strength

The authors used sufficiently large scale of dataset from 307 individuals of 41 specifies of Heliconiini butterflies to solidify the quantitative conclusions, and present new microscopy data for fine neuroanatomical comparison of the central complex.

Weakness

(1) Although the figures display a concise summary of anatomical findings, it would be difficult for non-experts to learn from this manuscript to identify the same neuronal processes in the raw confocal stacks. It would be helpful to have instructive movies to show step by step guide for identifications of neurons of interests, segmentations and 3D visualizations (rotation) for several examples including ER neurons (to supplement texts in line 347-353) and Allatostatin A neurons.

(2) Related to (1), it was difficult for me to access if the data in Fig 7 support the author's conclusions that ER neuron number increased in Heliconius Melpomene. By my understanding, the resolution of this dataset isn't high enough to trace individual axons and therefore authors do not rule out that the portion of "ER ring neurons" in Heliconius may not innervate the ER, as stated in Line 635 "Importantly, we also found that some ER neurons bypass the ellipsoid body and give rise to dense branches within distinct layers in the fan-shaped body (ER-FB)". If they don't innervate the ellipsoid body, why are they named as "ER neurons"?

(3) Discussions around the line 577-584 requires the assumption that each ellipsoid body (EB) ring neuron typically arborise in a single microglomerulus to form largely one-to-one connection with TuBu neurons within the bulb (BU), and therefore the number of BU microglomeruli should provide an estimation of the number of ER neurons. Explain this key assumption or provide an alternative explanation.

(4) The details of antibody information are missing in the Key resource table. Instead of citing papers, list the catalogue numbers and identifier for commercially available antibodies, and describe the antigen and if they are monoclonal or polyclonal. Are antigens conserved across species?

(5) I did not understand why authors assume that foraging to feed on pollens is more difficult cognitive task than foraging to feed on nectars. Would it be possible that they are equality demanding tasks but pollen feeding allows Heliconius to pass more proteins and nucleic acids to their offsprings and therefore they can develop larger mushroom bodies?

Comments on revisions:

The authors fully addressed my concerns and significantly improved the accessibility of the manuscript.

---

## [Referee Report · Reviewer #2 (Public review)]

Summary

In this study, Farnsworth et al. ask whether the previously established expansion of mushroom bodies in the pollen foraging Heliconius genus of Heliconiini butterflies co-evolved with adaptations in the central complex. Heliconius trap line foraging strategies to acquire pollen as a novel resource require advanced spatial memory mediated by larger mushroom bodies but the authors show that related navigation circuits in the central complex are highly conserved across the Heliconiini tribe, with a few interesting exceptions. Using general immunohistochemical stains and 3D reconstruction, the authors compared volumes of central complex regions and unlike the mushroom bodies, there was no evidence of expansion associated with pollen feeding. However, a second dataset of neuromodulator and neuropeptide antibody labeling reveal more subtle differences between pollen and non-pollen foragers and highlight sub-circuits that may mediate species-specific differences in behavior. Specifically, the authors found an expansion of GABAergic ER neurons projecting to the fan shaped body in Heliconius which may enhance their ability to path-integrate. They also found differences in Allatostatin A immunoreactivity, particularly increased expression in the noduli associated with pollen feeding. These differences warrant closer examination in future studies to determine their functional implication on navigation and foraging behaviors.

Strengths

The authors leveraged a large morphological data set from the Heliconiini to achieve excellent phylogenetic coverage across the tribe with 41 species represented. Their high quality histology resolves anatomical details to the level of specific, identifiable tracts and cell body clusters. They revealed differences at a circuit level, which would not be obvious from a volumetric comparison. The discussion of these adaptations in the context of central complex models is useful for generating new hypotheses for future studies on the function of ER-FB neurons and the role of Allatostatin A modulation in navigation.

The conclusions drawn in this paper are measured and supported by rigorous statistics and evidence from micrographs.

Weaknesses

The majority of results in this study do not reveal adaptations in the central complex associated with pollen foraging. However, reporting conserved traits is useful and illustrates where developmental or functional constraints may be acting. The authors have now revised the introduction to set up two alternate hypotheses..

In the main text, the authors describe differences in GABAergic ER neurons between H. melpomene and an outgroup species, with additional images from other species in Figure S4. Quantification of ER cells in these other species would strengthen the claim that these are increased in Heliconius and not just the focal species, but this may hopefully be pursued in future studies.

Comments on revisions:

I am satisfied with the authors' revisions.

---

## [Author Response]

The following is the authors’ response to the original reviews.

**eLife Assessment**
The analysis of neural morphology across Heliconiini butterfly species revealed brain area specific changes associated with new foraging behaviours. While the volume of the centre for learning and memory, the mushroom bodies, was known to vary widely across species, new, valuable results show conservation of the volume of a center for navigation, the central complex. The presented evidence is convincing for both volumetric conservation in the central complex and fine neuroanatomical differences associated with pollen feeding, delivered by experimental approaches that are applicable to other insect species. This work will be of interest to evolutionary biologists, entomologists, and neuroscientists.

Many thanks for your assessment and time handling this manuscript. We value the constructive input of both reviewers and believe that the result is an improved publication.

**Public Reviews:**

**Reviewer #1 (Public review):**
The authors previously reported that Heliconius, one genus of the Heliconiini butterflies, evolved to be efficient foragers to feed pollen of specific plants and have massively expanded mushroom bodies. Using the same image dataset, the authors segmented the central complex and associated brain regions and found that the volume of the central complex relative to the rest of the brain is largely conserved across the Heliconiini butterflies. By performing immunostaining to label a specific subset of neurons, the authors found several potential sites of evolutionary divergence in the central complex neural circuits, including the number of GABAergic ellipsoid body ring neurons and the innervation patterns of Allatostatin A expressing neurons in the noduli. These neuroanatomical data will be helpful to guide future studies to understand the evolution of the neural circuits for vector-based navigation.

We thank Reviewer 1 for the constructive feedback and criticism, which will have strengthened this publication.

Strengths:The authors used a sufficiently large scale of dataset from 307 individuals of 41 species of Heliconiini butterflies to solidify the quantitative conclusions and present new microscopy data for fine neuroanatomical comparison of the central complex.Weaknesses:(1) Although the figures display a concise summary of anatomical findings, it would be difficult for non-experts to learn from this manuscript to identify the same neuronal processes in the raw confocal stacks. It would be helpful to have instructive movies to show a step-by-step guide for identification of neurons of interest, segmentations, and 3D visualizations (rotation) for several examples, including ER neurons (to supplement texts in line 347-353) and Allatostatin A neurons.

We approached this with the following logic:

All 3D segmentations were animated, to illustrate how they are generated from raw imaging data. This means we are providing a video file for each major species group (*Heliconius*/outgroup-Heliconiini) for Figure 4 (general CX anatomy), Figure 7 (ER neuron projections), Figure S5 (ER neuron/bulb anatomy). This visual connection should help the reader relate 3D segmentations to image stacks. We have also added a reference to these videos in the relevant Figure captions.

We also annotated image stacks, but did so selectively. We annotated key stacks of Figure 4 (general CX anatomy), Figure 7 (ER neuron projections), Figure S5 (ER neuron/bulb anatomy) and include a reference in figure caption to them.

We refrained from annotating stacks of Figures 5, 6, 8 and S4. This is because we believe that the annotations we have performed in the figure panels will be sufficient for readers interested in the finer detail of these anatomies who are familiar with general CX anatomy.

We believe that our approach will help the reader to gain a visual illustration of those parts of the manuscript which report key results and novel insights, such as ER neuronal variation, and that the data and figures collectively provide accessible information sufficient for this purpose.

Text changes in Figure captions 4, 7 and S5: “See animated 3D segmentations and annotated stacks in file repository.”

(2) Related to (1), it was difficult for me to assess if the data in Figure 7 support the author's conclusions that ER neuron number increased in Heliconius Melpomene. By my understanding, the resolution of this dataset isn't high enough to trace individual axons and therefore authors do not rule out that the portion of "ER ring neurons" in Heliconius may not innervate the ER, as stated in Line 635 "Importantly, we also found that some ER neurons bypass the ellipsoid body and give rise to dense branches within distinct layers in the fan-shaped body (ER-FB)". If they don't innervate the ellipsoid body, why are they named as "ER neurons"?

Thanks for pointing to this. We believe this is primarily a nomenclature issue but have tried to specify in the text.

Ultimately, neurons from this group that project to the EB forming the actual ring neurons and those that project to the FB with unclear function, thus far, emerge through the same lineage, DALv2 (as determined by Kandimalla et al 2023) and therefore have common developmental origin (also noted by Homberg et al 2018). To acknowledge their common developmental origin and to simplify nomenclature, and therefore also provide easier comprehension by non-experts, we specify which DALv2 progeny project to which areas, but refer to both adult neuron populations to “ER neurons”. We have changed the following text to acknowledge our definition specifically, which we hope mitigates the understandable confusion.

Lines 354-357: “Here, we refer to these neurons, as well as those neurons projecting to the fan-shaped body (GU neurons in [66]), as ER neurons due to their common developmental origin [45,66] and to simplify anatomical descriptions.”

Lines 386-387: “Whether these ER neurons solely branch in the fan-shaped body, as shown for GU neurons elsewhere [66] or have additional side branches entering the ellipsoid body is not clear.”

(3) Discussions around the lines 577-584 require the assumption that each ellipsoid body (EB) ring neuron typically arborises in a single microglomerulus to form a largely one-to-one connection with TuBu neurons within the bulb (BU), and therefore, the number of BU microglomeruli should provide an estimation of the number of ER neurons. Explain this key assumption or provide an alternative explanation.

Thanks for this. We do not think that our hypothesis necessarily requires any specific assumptions regarding the ratio of microglomerulus to ER or TuBu neurons. Even in *Drosophila* the ratio of ER to MG is only approximately 1:1, as some microglomeruli seem to combine into one. In other species this relationship might be very different. Indeed, our data suggests that in outgroup-Heliconiini the ratio is 4.4 microglomeruli to 1 ER neuron, and in *Heliconius* it is 3.4. However, as these MG numbers are extrapolated and cannot be precisely counted, they may be too imprecise to come to a definite conclusion, hence why we do not mention this in the text. Importantly, extrapolation in the current form is a valid additional way for us to describe overall bulb anatomy (next to bulb volume, average microglomerulus size).

In any case, the inference we make here is that a conserved bulb anatomy in volume, MG numbers and size supports our assumption that the additional neurons in the ER neuron group/DALv2 progeny do not arborize in the bulb, but do so in the SMP/SLP region and in the fanshaped body. We believe we have described this inference accurately in the current manuscript.

An additional point, not mentioned in the manuscript, but emerging through lineage annotations of connectome data, is that some DALv2 progeny have been identified as MBONs as well as being GABA-ergic, which could potentially be the ER-FB neurons that we describe (Schlegel et al 2024 Nature). We refrain from mentioning this here, as its too speculatory, but we thought the reviewer may be interested in this observation.

(4) The details of antibody information are missing in the Key resource table. Instead of citing papers, list the catalogue numbers and identifier for commercially available antibodies, and describe the antigen, and whether they are monoclonal or polyclonal. Are antigens conserved across species?

We have now added substantial information to Table 2, including research resource identifiers (RRIDs) and antigen descriptions, as well as information about specificity and conservation. In the text itself, in line 757, we already provide publications that have illustrated conservation very extensively.

We believe that with the additional information provided in Table 2, all necessary information is now provided.

(5) I did not understand why authors assume that foraging to feed on pollens is a more difficult cognitive task than foraging to feed on nectar. Would it be possible that they are equally demanding tasks, but pollen feeding allows Heliconius to pass more proteins and nucleic acids to their offspring and therefore they can develop larger mushroom bodies?

This is an excellent point. Our current understanding is that pollen feeding is a cognitively more demanding task, because, (a) the density of pollen resources is lower than nectar resources, and (b) the competition for pollen is higher (pollen is depleted quickly, and *Heliconius* compete with each other, and other taxa including hummingbirds). There is therefore a benefit to high foraging efficiency, which favours the evolution of learning. This is likely reinforced by the long lives of *Heliconius* which live up to a year, compared to ~4 weeks for most outgroups and the temporal stability of major pollen resources, resulting in a memorised location providing benefit for the long periods of time (Young and Montgomery 2020 Proc B).

We now refer to an additional publication (Young and Montgomery 2020 Proc B) in lines 103-104 for a fuller description of the ecology of pollen feeding, and in the current manuscript simply focus on the impact of mushroom body expansion on the CX.

**Reviewer #2 (Public review):**
Summary:In this study, Farnsworth et al. ask whether the previously established expansion of mushroom bodies in the pollen foraging Heliconius genus of Heliconiini butterflies co-evolved with adaptations in the central complex. Heliconius trap line foraging strategies to acquire pollen as a novel resource require advanced spatial memory mediated by larger mushroom bodies, but the authors show that related navigation circuits in the central complex are highly conserved across the Heliconiini tribe, with a few interesting exceptions. Using general immunohistochemical stains and 3D reconstruction, the authors compared volumes of central complex regions, and unlike the mushroom bodies, there was no evidence of expansion associated with pollen feeding. However, a second dataset of neuromodulator and neuropeptide antibody labeling reveals more subtle differences between pollen and non-pollen foragers and highlights sub-circuits that may mediate species-specific differences in behavior. Specifically, the authors found an expansion of GABAergic ER neurons projecting to the fanshaped body in Heliconius, which may enhance their ability to path-integrate. They also found differences in Allatostatin A immunoreactivity, particularly increased expression in the noduli associated with pollen feeding. These differences warrant closer examination in future studies to determine their functional implication on navigation and foraging behaviors.

We thank Reviewer 2 for the constructive and thorough review. We believe that addressing these criticisms will have improved this publication.

Strengths:The authors leveraged a large morphological data set from the Heliconiini to achieve excellent phylogenetic coverage across the tribe with 41 species represented. Their high-quality histology resolves anatomical details to the level of specific, identifiable tracts and cell body clusters. They revealed differences at a circuit level, which would not be obvious from a volumetric comparison. The discussion of these adaptations in the context of central complex models is useful for generating new hypotheses for future studies on the function of ER-FB neurons and the role of Allatostatin A modulation in navigation.The conclusions drawn in this paper are measured and supported by rigorous statistics and evidence from micrographs.Weaknesses:The majority of results in this study do not reveal adaptations in the central complex associated with pollen foraging. However, reporting conserved traits is useful and illustrates where developmental or functional constraints may be acting. The implied hypothesis in the introduction is that expansion of mushroom bodies in Heliconius co-evolved with central complex adaptations, so it may be helpful to set up the alternate hypotheses in the beginning.

Thank you for this relevant comment. We have added to the text in lines 124-128, as follows

“Indeed, these circumstances permit us to test the hypotheses that modifications in the mushroom bodies either occurred in isolation from other integrative centres, or that they occurred in concert with specific changes in centres, such as the central complex. This provides insights into the functional flexibility of two interacting, integrative centres across evolutionary time.”

In the main text, the authors describe differences in GABAergic neurons "across several species" but only one Heliconius and one outgroup species seem to be represented in the figures. ER numbers in Figure 7H are only compared for these two species. If this data is available for other species, it would strengthen the paper to add them to the analysis, since this was one of the most intriguing findings in the study. I would want to know if the increased ER number is a trend in Heliconius or specific to H. melpomene.

This points to imprecise phrasing. We indeed have additional data in other species, but unfortunately not to an extent that would permit quantification of cell numbers, which is why we chose to put these data into the supplement, Fig. S4.

We modified the text to more directly point at the additional data in Fig S4, now reading in lines 362-368

“…, we noticed a pronounced difference in a portion of projections leading into the fan-shaped body and a strong difference in signal inside layer III in our two focal species *H. Melpomene* and *D. iulia*, as well as other representatives of the Heliconiini tribe (Figure S4A-B, Figure 7). To understand how these differences could have occurred, we quantified ER neuron numbers in our focal species, and identified a significant difference, reflecting a 35% increase in Heliconius (*t* = 4.221, *P* = 0.004; Figure 7H).”

**Recommendations for the authors:**

**Reviewer #1 (Recommendations for the authors):**
(1) Add a detailed description about each of the tiff files that were deposited at https://doi.org/10.5281/zenodo.15304965. It was hard for me to relate these raw images with the Figure panels. For instance, "Melp_GAD_26-F_detailed_conc.tif" in the Figure 7 folder seems to be used to make Figure 7L and N, but that information is cryptic.

We agree with the reviewer. We added further descriptions, and have created a detailed readme file which explains which original file refers to which figure. Together with the efforts for Reviewer 1’s first comment, we hope that this updated version of our repository is easier to understand.

In addition, we made additional changes in image orientation in some of the files supplied, and which were originally incorrect.

(2) Add descriptions about the dataset for large-scale volumetric analysis. With the current methods and texts, it is hard to understand what kinds of staining and microscopes were used. I initially thought that they could be micro-CT data.

We have made two improvements:

We have added an additional readme file to explain the different datasets, and which datasets were used for each figure, to relate them to the original data deposited at zenodo.org (see your previous comment).

We have added descriptions in several places in the manuscript file, i.e.

Lines 133-135, now reading “To assess evidence of volumetric changes in the central complex and associated neuropils, we drew data from a large dataset of immunostained brains from 307 individuals of 41 species, …”

Lines 144-149, now reading “We used a combination of phylogenetic comparative analysis across a large dataset of brains immunostained against the structural marker synapsin in 41 species and 307 individuals, and more targeted sampling of species that represent the behavioural and neuroanatomical diversity of Heliconiini for more fine-scale assessments of patterns of divergence in substructures of the CX with various antibodies (Figure 1A-B).”

(3) Line 275: Non-expert readers would need an explanation about what the gamma lobe is.

Agreed and added in line 273

“Some of the ventral projections seemed to directly originate from the γ lobe, a portion of the mushroom body, thus potentially labelling projections of mushroom body output neurons into the fan-shaped body (Figure 5a-c) [12,21].”

(4) Figures 4 I-L are missing.

We modified the figure caption accordingly, and address annotated differences more directly. This section now reads

“G/H: Labelling reveals two distinguishable layers in the fan-shaped body while additional staining elsewhere reveals further detail (arrows in G/H-2/3). Thicker tract conflations indicate the columnar architecture determined through the four columnar neuron bundles (arrowheads in G/H-3). Labelling in the EB reveals two pronounced layers (arrows in G/H-1/2), while obvious columns could not be indicated. PB protocerebral bridge, FB fan-shaped body, EB ellipsoid body. A anterior, P posterior. Scale bars are 50 μm.”

(5) In the current version of Figure 1B, AOTU is displayed with the mushroom body. The authors can emphasize its relation to the central complex by showing it on the right side of panels together with the central complex.

Great suggestion. We have done this now. We have kept the AOTU at the scale of the MB, indicated by the different scale bars of the bottom of the figure, as we’re showing the CX at a slightly larger scale.

(6) Figure 1C: What do the colors of the lines represent?

We now changed these colours so that they correspond to the colours chosen in Figures 2 and S2 as well as in a previous publication of the lab, added an asterisk next to Heliconius aoede, and added text to the figure legend:

“Colour indicates focal groups here and elsewhere [29]. The asterisk at the branch of *H. aoede* indicates a secondary loss of pollen feeding.”

(7) Figures 2A and B: What does the size of the circles represent? I guess that small ones are individuals, and larger ones are species averages. Plots with only species averages would be easier to see. It is difficult to distinguish Heliconius and Helicononius aoede in these panels. It would be easier if Heliconius circles were outlined with thin black lines.

Thanks for this. We wanted to keep both the averages and individual data points in one figure, as to not overcrowd the manuscript with additional figures. We still hope that the changes we made address the confusion sufficiently. We made the following modifications to Figure 2 and S1 and S2:

(1) Added text in the figure legend clarifying what solid and transparent circles indicate (“Solid data points indicate species averages, while opaque circles indicate individual data points.”)

(2) Added, as suggested, additional contours, to all *Heliconius* data points, and added corresponding text to the legend (“Black contours indicate Heliconius sp. data points.”)

(3) Changed opacity settings of individual data points.

**Reviewer #2 (Recommendations for the authors):**
(1) Line 391 and Methods. It was unclear how the extrapolated microglomeruli numbers were calculated. Please clarify this in the methods.

Agreed. We substantially modified the text to address this.

Lines 392-396: “We generated high resolution images of the bulb to determine its size (Figure S5 C-F), and 3D segmented seven microglomeruli per individual with which we generated an extrapolated approximation of total microglomeruli number by dividing bulb volume with average microglomerulus volume. This was necessary as most microglomeruli were not discernible from each other (Figure S5 G-H).”

Lines 862-873: “To segment the bulb, we created high resolution images and were particularly careful to only segment the area of the bulb that comprised large synapses/glomeruli, excluding parts of the LEa/IT projection. This was essential, because we relied on extrapolating the total number of microglomeruli from a subset of segmented microglomeruli and the total volume that contained microglomeruli, which means any section containing tracts and not glomerular structures would skew the estimated total number of microglomeruli. Extrapolation was necessary, as not all microglomeruli were visually discernible. We achieved an unskewed bulb volume by leaving out dense pieces of tubulin-positive tract material. We segmented seven microglomeruli per individual from the posterior section of the bulb, where they were most clearly visible, to get the most comparable impression across individuals and species. We then calculated average microglomerulus size and divided this by bulb volume to determine an approximation of microglomeruli number.”

(2) Line 439. It would be helpful to add that Kaiser et al. studied honeybees.

Agreed! Now reads in lines 443-444

“Moreover, Kaiser et al. [75] identified Allatostatin A expression in three fan-shaped and two ellipsoid body layers in the honey bee brain, …”

(3) Line 492. "outcome" should be "outcomes".

We believe that this refers to original line 481. Corrected. Thank you.

(4) Figure 3B. If there is significance to the colors and triangle directions, please include a key/legend.

We have added:

“Cell type depictions are examples with localisation inside each neuropil being purely visual (as well as their colour), while triangles indicate approximate output sites.”

We also corrected the following issues that were noted during our revisions:

line 587, wrong reference.

We updated references 37 and 44, which are now respectively

Hodge, E. A. et al. Modality-specific long-term memory enhancement in Heliconius butterflies. Philos Trans R Soc Lond B Biol Sci 380, 20240119 (2025).

Hodge, E. A. et al. Conservation of sensory pathways implies a localised change in the mushroom bodies is associated with cognitive evolution in Heliconius butterflies. Evol qpag005 (2026) doi:10.1093/evolut/qpag005.

Figure S5 had an error in panels C and D, where the pictures in C were actually for *H. Melpomene* in D and the reverse; the other panels were correct. We have corrected this.

In the data submitted on Zenodo: we corrected a few inconsistencies in channel colours and orientation in the .tiff files for Fig 6, 8 and S4.

We added important bulb 3D segmentation files to the repository on Zenodo.